# Decentralized Nonconvex Optimization under Heavy-Tailed Noise: Normalization and Optimal Convergence

**Shuhua Yu**
Department of Electrical and Computer Engineering
Carnegie Mellon University
Pittsburgh, PA 15213, USA
`shuhuay@andrew.cmu.edu`

**Dušan Jakovetić**
Department of Mathematics and Informatics
University of Novi Sad
Novi Sad, 21000, Serbia
`dusan.jakovetic@dmi.uns.ac.rs`

**Soummya Kar**
Department of Electrical and Computer Engineering
Carnegie Mellon University
Pittsburgh, PA 15213, USA
`soummyak@andrew.cmu.edu`

## Abstract

Heavy-tailed noise in nonconvex stochastic optimization has garnered increasing research interest, as empirical studies, including those on training attention models, suggest it is a more realistic gradient noise condition. This paper studies first-order nonconvex stochastic optimization under heavy-tailed gradient noise in a decentralized setup, where each node can only communicate with its direct neighbors in a predefined graph. Specifically, we consider a class of heavy-tailed gradient noise that is zero-mean and has only $p$-th moment for $p \in (1, 2]$. We propose `GT-NSGDm`, Gradient Tracking based Normalized Stochastic Gradient Descent with momentum, that utilizes normalization, in conjunction with gradient tracking and momentum, to cope with heavy-tailed noise on distributed nodes. We show that, when the communication graph admits primitive and doubly stochastic weights, `GT-NSGDm` guarantees, for the *first* time in the literature, that the expected gradient norm converges at an *optimal non-asymptotic rate* $O\big(1/T^{(p-1)/(3p-2)}\big)$, which matches the lower bound in the centralized setup. When the tail index $p$ is unknown, `GT-NSGDm` attains a non-asymptotic rate $O\big(1/T^{(p-1)/(2p)}\big)$ that is, for $p < 2$, topology independent and has a speedup factor $n^{1-1/p}$ in terms of the number of nodes $n$. Finally, experiments on nonconvex linear regression with tokenized synthetic data and decentralized training of language models on a real-world corpus demonstrate that `GT-NSGDm` is more robust and efficient than baselines.

## 1 Introduction

In this paper, we address the problem of nonconvex stochastic optimization under heavy-tailed gradient noise in the decentralized setup. Consider a graph with $n$ nodes connected by a predefined topology $\mathcal{G} := (\mathcal{V}, \mathcal{E})$, where $\mathcal{V} := \{1, \dots, n\}$ is the set of node indices, and $\mathcal{E}$ is the collection of directed pairs $(i, r)$, $i, r \in \mathcal{V}$ such that node $i$ can send information to the neighboring node $r$. Each node $i \in \mathcal{V}$ holds a local nonconvex differentiable cost function $f_i : \mathbb{R}^d \to \mathbb{R}$, and can access its stochastic gradient, subject to zero mean noise with a bounded $p$-th moment for some $p \in (1, 2]$. Cooperatively, these nodes aim to solve $\min_{\boldsymbol{x} \in \mathbb{R}^d} f(\boldsymbol{x}) := (1/n) \sum_{i=1}^{n} f_i(\boldsymbol{x})$, through local computation and peer-to-peer communication only with their immediate neighbors.

Decentralized optimization in the above formulation has been studied for decades (Tsitsiklis et al., 1986), and has recently attracted growing research interest due to its advantages in scalability and privacy preservation across a wide range of distributed machine learning, signal processing, and

control tasks over networks (Nedić et al., 2018; Li et al., 2020; Kairouz et al., 2021). For instance, in privacy-sensitive applications such as those in the medical domain (Brisimi et al., 2018), training data are often distributed across $n$ nodes due to privacy constraints. In such cases, each $f_i$ represents an empirical risk function, e.g., a neural network, defined over the local dataset on node $i$, and all nodes collaboratively train a global predictive model via peer-to-peer communication without sharing raw data. Moreover, decentralized optimization is also employed in data centers to reduce communication bottlenecks associated with the central node in traditional centralized training paradigms (Lian et al., 2017).

In decentralized optimization, first-order methods are widely favored for their simplicity and scalability (Xin et al., 2020b). However, computing the exact gradient of each local objective function $f_i$ at every iteration can be computationally expensive, particularly in large-scale settings where each node holds a substantial volume of local data. To alleviate this computational burden, decentralized stochastic gradient methods, which approximate exact gradients, have been extensively studied. Most existing approaches, including decentralized stochastic (sub)gradient descent (Sundhar Ram et al., 2010; Koloskova et al., 2020; Wang & Joshi, 2021), variance reduction techniques (Yuan et al., 2018), and gradient tracking-based schemes (Di Lorenzo & Scutari, 2016; Pu & Nedić, 2021), typically assume that stochastic gradient noise has a *finite variance*. Nevertheless, recent empirical and theoretical evidence indicates that, when optimizing certain neural network architectures, especially attention-based models such as Transformers (Vaswani et al., 2017), the gradient noise often follows a *heavy-tailed distribution*[1] with significantly large or even *infinite variance* (Simsekli et al., 2019; Zhang et al., 2020; Gorbunov et al., 2020; Gurbuzbalaban et al., 2021; Ahn et al., 2024; Kunstner et al., 2024). The presence of heavy-tailed gradient noise poses substantial challenges for existing methods. Empirically, some stochastic gradient descent (SGD) based methods can suffer from instability and even dramatic drop of training accuracies (Zhang et al., 2020; Charles et al., 2021; Yang et al., 2022), particularly in distributed large-cohort training. Theoretically, unbounded variance renders many established analyses invalid, and in *centralized* settings it necessitates the use of nonlinear adaptive techniques such as clipping, sign, and normalization (Zhang et al., 2020; Sadiev et al., 2023; Compagnoni et al., 2025b; Hübler et al., 2024; Liu & Zhou, 2025; Armacki et al., 2025) to combat the strong noise. However, incorporating such adaptive strategies in *decentralized* algorithms introduces inherent *nonlinearity* into the algorithmic dynamics associated with the average-sum structured function $f$, making the design and analysis of decentralized algorithms under heavy-tailed noise significantly more challenging.

Decentralized optimization under heavy-tailed gradient noise remains underexplored. To the best of our knowledge, only recent studies Sun & Chen (2024); Yu et al. (2023) have attempted to address this problem under restrictive assumptions. Specifically, Sun & Chen (2024) considers zero-mean gradient noise with bounded $p$-th central moment ($p \in (1, 2]$) similar to our setting but assumes a *compact* domain or *bounded gradients*. Their proposed decentralized gradient descent method with $\ell_2$ gradient clipping achieves almost sure convergence for *strongly convex* local functions. However, the restrictive compact domain or gradients assumption in Sun & Chen (2024) limits its practical applicability, and the convergence rate is not explicitly provided. Another work, Yu et al. (2023), also assumes strongly convex local objectives and develops a decentralized gradient method with smoothed clipping and error feedback under gradient noise that is zero-mean, *symmetric*, and has bounded first absolute moment, showing an *in-expectation* convergence rate of $1/t^\delta$ for some $\delta \in (0, 2/5)$. Although the noise assumption in Yu et al. (2023) is weaker than ours (as it requires only a first-moment bound), the additional assumptions of *noise symmetry* and the dependence of the rate exponent $\delta$ on both the problem dimension and condition number restrict its general applicability. Moreover, both works Sun & Chen (2024); Yu et al. (2023) assume strong convexity, whereas many practical optimization problems involving heavy-tailed noise, particularly in modern machine learning, are inherently nonconvex. Further, the convergence rates in Sun & Chen (2024); Yu et al. (2023) are either unclear or sub-optimal, even compared to the optimal iteration complexity bound $O\big(1/T^{(p-1)/(3p-2)}\big)$ for general nonconvex functions. In this work, we relax these restrictive assumptions and address the following question:

---

[1]A random variable $X$ is called heavy-tailed if it exhibits a heavier tail than any exponential distribution; formally, for any constant $a > 0$, $\limsup_{x \to \infty} \mathbb{P}(X > x)e^{ax} = \infty$ (Nair et al., 2022). While some heavy-tailed distributions, such as log-normal and Weibull, still have bounded variance, this paper also considers the sub-class of heavy tailed gradient noise that may have unbounded (infinite) variance such as $\alpha$-stable noise.

*Can we design a decentralized algorithm for **nonconvex** optimization under general zero-mean gradient noise with only a finite p-th moment for $p \in (1,2]$ with **optimal iteration complexity**?*

## 1.1 Contributions

We answer this question affirmatively through the following key contributions:

- We develop a decentralized method, called `GT-NSGDm`, using normalization, coupled with momentum variance reduction, to combat heavy-tailed noise, and using gradient tracking to handle cross-node heterogeneity. To further shed light on the design of `GT-NSGDm`, we provide a negative result for a vanilla variant of normalized decentralized SGD that employs neither gradient tracking nor momentum.

- For general nonconvex and smooth local functions $f_i$'s that are bounded from below, we show that `GT-NSGDm` converges in expectation at a rate $O\big(1/T^{(p-1)/(3p-2)}\big)$, which matches the lower bound in the centralized setting and is order-optimal. Our convergence rate significantly improves upon related works (Sun & Chen, 2024; Yu et al., 2023), which assume strong convexity and lack an explicit rate exponent.

- When the tail index $p$ is unknown, `GT-NSGDm` achieves a rate of $O(1/T^{(p-1)/(2p)})$, matching the best-known rate in the centralized setting without requiring knowledge of $p$. Notably, for $p \in (1,2)$ and sufficiently large $T$, this rate is *independent* of the network topology and exhibits a *speedup* in the number of nodes, with a factor of $n^{1-1/p}$.

- We test our theoretical findings in nonconvex linear regression models on a synthetic dataset that is built to simulate language tokens under controlled heavy-tailed noise injections. We also test `GT-NSGDm` on distributed training of decoder-only Transformer models on Multi30k datasets (Elliott et al., 2016). Experiments on multiple variants of network topologies show that `GT-NSGDm` is more robust to injected and empirical heavy-tailed noise and converges faster.

## 1.2 Related Work

**Heavy-tailed gradient noise.** Recent empirical studies suggest that the distribution of gradient noise in training various deep learning models resembles heavy-tailed distributions, such as Lévy's $\alpha$-stable distribution (Simsekli et al., 2019; Zhang et al., 2020; Barsbey et al., 2021; Battash et al., 2024). For instance, the work Zhang et al. (2020) demonstrates that the empirical distribution of gradient norm samples during BERT pre-training closely aligns with an $\alpha$-stable distribution, rather than a Gaussian one (see their Figure 1). The presence of heavy-tailed gradient noise is also supported by theoretical insights (Simsekli et al., 2019; Peluchetti et al., 2020; Gurbuzbalaban et al., 2021; Barsbey et al., 2021). In particular, Simsekli et al. (2019) leverages generalized central limit theorems to show that the gradient noise in SGD can converge to an $\alpha$-stable random variable.

**Adaptive methods.** Under heavy-tailed noise, vanilla SGD based methods are shown to suffer from slower convergence or even model collapses in *centralized* settings (Zhang et al., 2020) as well as *distributed settings with a central server* (Yang et al., 2022; Lee et al., 2025), and adaptive methods such as clipping and normalization are introduced to stabilize training dynamics. In *centralized* settings, the work Zhang et al. (2020) provides lower bounds for both nonconvex and strongly convex smooth functions, showing that SGD with gradient clipping achieves *in-expectation* upper bounds matching lower bounds. In Sadiev et al. (2023); Liu et al. (2023); Nguyen et al. (2023); Chezhegov et al. (2024), the authors show that when equipped with gradient clipping, SGD, accelerated methods, AdaGrad (Duchi et al., 2011), and Adam (Kingma & Ba, 2014) can achieve (near-)optimal *high-probability* convergence under various function assumptions. Besides, the work Compagnoni et al. (2025b) shows that signSGD is also robust to heavy-tailed noise through the lens of stochastic differential equations. Further, SGD with gradient normalization, which advantageously requires less hyper-parameter tuning than clipping, is shown to achieve optimal *in-expectation* convergence (Hübler et al., 2024; Liu & Zhou, 2025; Sun et al., 2024). Our method incorporates the same normalization and variance reduction approach as Liu & Zhou (2025). Notably, in another line of works Jakovetić et al. (2023); Armacki et al. (2025; 2024), the authors conduct a unified convergence analysis for generic nonlinear methods including clipping, sign, and normalization under *symmetric* noise with positive probability mass around zero without assuming any noise moment bound

or only assuming a first absolute noise moment bound. In *distributed settings with a server*, the work Gorbunov et al. (2024) proposes an algorithm that incorporates an error feedback mechanism, wherein clipping is applied to the discrepancy between a local gradient estimator and a stochastic gradient, and establishes optimal *high-probability* bounds. Moreover, the work Compagnoni et al. (2025a) shows that distributed signSGD converges to an asymptotic neighborhood depending on the 'fatness' of the noise tail. When multiple local updates are permitted between communication rounds, the authors of Yang et al. (2022) show that clipping per local step achieves order-optimal in-expectation convergence, albeit under a restrictive *bounded gradient* assumption. In addition, the work Lee et al. (2024) rigorously shows that distributed training under heavy-tailed noise can collapse with unbounded gradients, and shows that client-side adaptive methods can stabilize training with bounded regrets in an online learning framework. More recently, the work Lee et al. (2025) introduces the TailOPT framework, which adaptively leverages gradient geometry by applying clipping operators during local updates on distributed nodes and utilizing adaptive optimizers for global updates at the server, achieving in-expectation sublinear convergence rates that are independent of the moment parameter $p$.

**Nonlinearities in decentralized optimization.** Extending existing methods that are robust to heavy-tailed noise, whether developed for *centralized* settings or *distributed settings with a server*, to decentralized environments is highly nontrivial, primarily due to the *nonlinearities* introduced to peer-to-peer communication. This difficulty is reflected in that existing decentralized methods incorporating nonlinear adaptive techniques for other purposes often impose restrictive conditions (Yu & Kar, 2023; Li & Chi, 2025). For example, to achieve differential privacy through gradient clipping, the work Li & Chi (2025) establishes convergence in decentralized setups under the assumption of either *bounded gradient* or a *stringent similarity* condition, namely $\|\nabla f_i(\boldsymbol{x}) - \nabla f(\boldsymbol{x})\| \leq (1/12)\|\nabla f(\boldsymbol{x})\|$ for all $i \in [n]$ and all $\boldsymbol{x}$. Similarly, to attain adversarial robustness against gradient attacks, the authors of Yu & Kar (2023) employ gradient clipping with momentum, assuming that all local functions are convex, share a *common minimizer*, and that $\sum_{i=1}^{n} f_i$ is strongly convex. Furthermore, Taheri & Thrampoulidis (2023) integrates gradient tracking with normalization to address a max-margin problem. While gradient tracking is employed to ensure similarity among local gradient estimators, the method guarantees convergence only to the direction of the optimal solution under deterministic gradients. In this work, we significantly relax these conditions and demonstrate the effective use of nonlinearity (specifically, normalization) in decentralized optimization, thereby motivating broader applications of nonlinear techniques in this setting.

## 1.3 NOTATION

We denote by $\mathbb{N}_+$, $\mathbb{R}$, $\mathbb{R}_+$ and $\mathbb{R}^d$, respectively, the set of positive natural numbers, real numbers, nonnegative real numbers, and the $d$-dimensional Euclidean space. We use lowercase normal letters for scalars, lowercase boldface letters for vectors, and uppercase boldface letters for matrices. Further, we denote by $\mathbf{1}_k$ and $\mathbf{0}_k$ the all-ones and all-zeros vectors of size $k$, respectively, and by $\boldsymbol{I}_k$ the $k \times k$ identity matrix. We let $\|\boldsymbol{x}\|$ denote the $\ell_2$ norm of $\boldsymbol{x}$, and $\|\boldsymbol{A}\|_2$ denote the operator norm of $\boldsymbol{A}$. For functions $p(t)$ and $q(t)$ in $t$, we write $p(t) = O(q(t))$ if $\limsup_{t \to \infty} p(t)/q(t) < \infty$. Finally, we use $\mathbb{E}$ to denote expectation over random quantities.

## 2 PROBLEM FORMULATION

We consider a graph with $n$ nodes, where each node holds a local and private function $f_i : \mathbb{R}^d \to \mathbb{R}$, and the nodes collectively minimize the unconstrained global objective $f(\boldsymbol{x}) := (1/n)\sum_{i=1}^{n} f_i(\boldsymbol{x})$ through peer-to-peer communication. We now present some standard assumptions on the problem.

**Assumption 1** (Finite lower bound). *There exists some $f_* := \inf_{\boldsymbol{x} \in \mathbb{R}^d} f(\boldsymbol{x}) > -\infty$.*

**Assumption 2** (*L*-smoothness). *The local function $f_i$ at each node $i \in [n]$ is differentiable and L-smooth, i.e., $\forall \boldsymbol{x}, \boldsymbol{y} \in \mathbb{R}^d, \|\nabla f_i(\boldsymbol{x}) - \nabla f_i(\boldsymbol{y})\| \leq L\|\boldsymbol{x} - \boldsymbol{y}\|$.*

We next introduce the heavy-tailed noise model. For each node $i \in \mathcal{V}$, at $t$-th iteration with query $\boldsymbol{x}_i^t$, the stochastic first-order oracle returns the gradient estimator $\boldsymbol{g}_i(\boldsymbol{x}_i^t, \boldsymbol{\xi}_i^t)$, where $\boldsymbol{\xi}_i^t$ denotes the random sample. Let $\Omega, \emptyset$ denote the universe, empty set, respectively. We use the following natural filtration, i.e., an increasing family of sub-$\sigma$-algebras, to denote the past history up to iteration $t$:

$$\mathcal{F}_{-1} := \{\Omega, \emptyset\}, \quad \mathcal{F}_t := \sigma\left(\{\boldsymbol{\xi}_i^0, \ldots, \boldsymbol{\xi}_i^{t-1} : i \in [n]\}\right), \forall t \geq 0.$$

We then assume this stochastic first-order oracle has the following properties.

**Assumption 3** (Heavy-tailed noise). *For any $\mathcal{F}_t$-measurable random vectors $\boldsymbol{x} \in \mathbb{R}^d$, we have the following: $\forall i \in [n], \forall t \geq 0$, (1) $\mathbb{E}[\boldsymbol{g}_i(\boldsymbol{x}, \boldsymbol{\xi}_i^t) \mid \mathcal{F}_t] = \nabla f_i(\boldsymbol{x})$; (2) There exist $p \in (1, 2]$, some constant $\sigma \geq 0$ such that $\mathbb{E}[\|\boldsymbol{g}_i(\boldsymbol{x}, \boldsymbol{\xi}_i^t) - \nabla f_i(\boldsymbol{x})\|^p \mid \mathcal{F}_t] \leq \sigma^p$; (3) The family $\{\boldsymbol{\xi}_i^t : \forall t \geq 0, i \in [n]\}$ of random samples is independent.*

*Remark* 1 (Heavy-tailed distributions). Assumption 3 covers a broad class of heavy-tailed distributions, including Lévy's $\alpha$-stable distributions, Student's $t$-distributions, and Pareto distributions. Note that we do not assume noise symmetry as in Yu et al. (2023), and when $p = 2$, Assumption 3 reduces to the standard bounded variance condition commonly assumed in the literature.

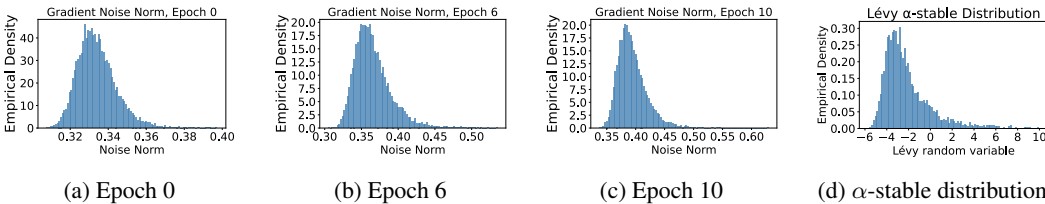

| (a) Epoch 0 | (b) Epoch 6 | (c) Epoch 10 | (d) $\alpha$-stable distribution |

Figure 1: Comparisons of the empirical density of gradient noise norm in different epochs of training a Transformer model with a synthetic Lévy $\alpha$-stable distribution.

*Remark* 2 (Empirical evidence). Similar to Zhang et al. (2020); Yang et al. (2022), we investigate the empirical distribution of the gradient noise norm $\|\boldsymbol{g}(\boldsymbol{x}, \boldsymbol{\xi}) - \nabla f(\boldsymbol{x})\|$ in a centralized setting by training a GPT model (Radford et al., 2018) with 3M parameters on the Multi30k dataset (Elliott et al., 2016), where $\boldsymbol{g}(\boldsymbol{x})$ denotes the mini-batch stochastic gradient and $\nabla f(\boldsymbol{x})$ denotes the full-batch gradient. We train the model for 12 epochs using SGD and plot the empirical density of the noise norm at the beginning of epochs 0, 6, and 10. As shown in Figure 1, as training progresses, the tail of the empirical gradient noise norm distribution becomes heavier (and longer) and increasingly resembles that of a synthetic $\alpha$-stable distribution.

For peer-to-peer communication in decentralized settings, we need to specify a mixing matrix $\boldsymbol{W}$ on graph $\mathcal{G} = (\mathcal{V}, \mathcal{E})$.

**Assumption 4** (Weight matrix). *The nonnegative weight matrix $\boldsymbol{W}$, whose $(i, r)$-th component of $\boldsymbol{W}$, denoted as $w_{ir}$, is positive if and only if $(i, r) \in \mathcal{E}$ or $i = r$, is primitive and doubly stochastic, i.e., $\mathbf{1}_n \boldsymbol{W} = \mathbf{1}_n$ and $\boldsymbol{W} \mathbf{1}_n = \mathbf{1}_n$.*

Assumption 4 is standard in the decentralized optimization literature (Xin et al., 2020a), and it guarantees that there exists some nonnegative $\lambda$, i.e., spectral gap, such that

$$\|\boldsymbol{W} - \mathbf{1}_n \mathbf{1}^\top / n\|_2 := \lambda < 1.$$

The assumed weight matrix $\boldsymbol{W}$ can be constructed on undirected and connected graphs (Olshevsky, 2014), and also on some directed and strongly connected graphs that are weight-balanced (Gharesifard & Cortés, 2012). For instance, the family of directed exponential graphs, is weight-balanced and serves as an important topology configuration in decentralized training (Assran et al., 2019).

## 3    ALGORITHM DEVELOPMENT: GT-NSGDM

We now describe the proposed Algorithm GT-NSGDm and discuss the intuition of its construction. We use $\boldsymbol{x}_i^t$ to denote the estimate of a stationary point for the global cost function $f$ at node $i$ and $t$-th iteration, and recall that $\boldsymbol{g}_i(\boldsymbol{x}_i^t, \boldsymbol{\xi}_i^t)$ denotes the corresponding stochastic gradient returned from a local first-order oracle. Motivated by the error-feedback approach in Yu et al. (2023), which serves as a momentum-type of variance reduction after applying a nonlinear operator to handle heavy-tailed noise, we also employ local momentum variance reduction

$$\boldsymbol{v}_i^t = \beta \boldsymbol{v}_i^{t-1} + (1 - \beta) \boldsymbol{g}_i(\boldsymbol{x}_i^t, \boldsymbol{\xi}_i^t), \tag{1}$$

where $\beta \in [0, 1)$ serves as the momentum coefficient. Then, we use gradient tracking (Di Lorenzo & Scutari, 2016) to handle heterogeneous local functions $\{f_i\}_{i=1}^n$. Specifically, we use an estimator

$\boldsymbol{y}_i^t$ to track the global gradient

$$\boldsymbol{y}_i^t = \sum_{r=1}^{n} w_{ir}\big(\boldsymbol{y}_r^{t-1} + \boldsymbol{v}_r^t - \boldsymbol{v}_r^{t-1}\big). \tag{2}$$

It is known that gradient tracking helps eliminate the dependence on heterogeneity among local functions $\{f_i\}_{i=1}^n$, such as the requirement of bounded gradient similarity. Furthermore, similar to the approach in Liu & Zhou (2025), which uses normalization to address heavy-tailed noise in *centralized* settings, we avoid applying normalization in the recursive updates of the local gradient estimator $\boldsymbol{v}_i^t$ in (1) and the global gradient tracker $\boldsymbol{y}_i^t$. Instead, normalization is applied only during the update of $\boldsymbol{x}_i^t$, with step size $\alpha$, and nonnegative mixing weights $\{w_{ir}\}$ where $w_{ir} > 0$ only when $(i, r) \in \mathcal{E}$ or $i = r$,

$$\boldsymbol{x}_i^{t+1} = \sum_{r=1}^{n} w_{ir}\Big(\boldsymbol{x}_r^t - \alpha \frac{\boldsymbol{y}_r^t}{\|\boldsymbol{y}_r^t\|}\Big). \tag{3}$$

We combine the local updates (1)(2)(3) on node $i \in \mathcal{V}$ and call it `GT-NSGDm`, Gradient Tracking based Normalized Stochastic Gradient Descent with momentum. When taking $\beta = 0$, this simplifies to momentum-free gradient tracking with normalization in step 3. However, our analysis shows that `GT-NSGDm` performs optimally for some $\beta \in (0, 1)$, making `GT-NSGDm` a non-trivial and optimal algorithmic design for the considered problem class. We provide a tabular description for `GT-NSGDm` in Algorithm 1, where all $\{\boldsymbol{x}_i^0\}$ are initialized from the same point $\bar{\boldsymbol{x}}^0$ for simplicity.

---

**Algorithm 1** `GT-NSGDm` at each node $i$

---

**Require:** $\boldsymbol{x}_i^{-1} = \boldsymbol{x}_i^0 = \bar{\boldsymbol{x}}^0, \boldsymbol{v}_i^{-1} = \boldsymbol{y}_i^{-1} = \mathbf{0}_d, \alpha, \beta, \{w_{ir}\}, T$.
1: **for** $t = 0$ to $T - 1$ **do**
2:      Sample $\boldsymbol{\xi}_i^t$;                                           (random sample for stochastic gradient)
3:      $\boldsymbol{v}_i^t \leftarrow \beta \boldsymbol{v}_i^{t-1} + (1 - \beta)\boldsymbol{g}_i(\boldsymbol{x}_i^t, \boldsymbol{\xi}_i^t)$;                       (local gradient estimator)
4:      $\boldsymbol{y}_i^t \leftarrow \sum_{r=1}^{n} w_{ir}(\boldsymbol{y}_r^{t-1} + \boldsymbol{v}_r^t - \boldsymbol{v}_r^{t-1})$;                   (local gradient tracker)
5:      $\boldsymbol{x}_i^{t+1} \leftarrow \sum_{r=1}^{n} w_{ir}\big(\boldsymbol{x}_r^t - \alpha \frac{\boldsymbol{y}_r^t}{\|\boldsymbol{y}_r^t\|}\big)$;               (peer-to-peer communication)
6: **end for**

---

*Remark* 3 (Why vanilla gradient normalization fails?). Although vanilla normalization is successfully used in *centralized* settings to robustify SGD against heavy-tailed noise (Hübler et al., 2024), its direct extension to the *decentralized* settings fails. Suppose we run a vanilla decentralized normalized (noiseless) gradient descent, i.e., in parallel $\forall i \in \mathcal{V}$,

$$\boldsymbol{x}_i^{t+1} = \sum_{i=1}^{n} w_{ir}\big(\boldsymbol{x}_r^t - \alpha \frac{\nabla f_r(\boldsymbol{x}_r^t)}{\|\nabla f_r(\boldsymbol{x}_r^t)\|}\big). \tag{4}$$

Then, the global average $\bar{\boldsymbol{x}}^t$ would update in the negative direction of *the sum of normalized local gradients*: $\bar{\boldsymbol{x}}^{t+1} = \bar{\boldsymbol{x}}^t - \frac{\alpha}{n} \sum_{r=1}^{n} \frac{\nabla f_r(\boldsymbol{x}_r^t)}{\|\nabla f_r(\boldsymbol{x}_r^t)\|}$. Let for some $t, \forall r \in \mathcal{V}, \boldsymbol{x}_r^t = \boldsymbol{x}_* = \arg\min \sum_{i=1}^n f_i(\boldsymbol{x})$, i.e., all nodes hold the optimal global solution that $\sum_{r=1}^{n} \nabla f_r(\boldsymbol{x}_*) = \mathbf{0}$. Since $\|\nabla f_r(\boldsymbol{x}_*)\|$ can be different quantities for $r = 1, \ldots, n$, due to function heterogeneity, then $\bar{\boldsymbol{x}}^{t+1}$ will move away from $\boldsymbol{x}_*$. Therefore, vanilla gradient normalization adds some intrinsic errors from heterogeneous local normalizations. By incorporating gradient tracking, we expect that $\boldsymbol{y}_r^t$ would converge to its global average $\bar{\boldsymbol{y}}^t$, and $\bar{\boldsymbol{y}}^t$ would converge to $(1/n) \sum_{i=r}^{n} \nabla f_i(\boldsymbol{x}_r^t)$. In this way, $\boldsymbol{x}_r^t$ would move along the direction of the *normalized sum of local gradients*, and thus emulating the centralized setting.

In the following claim, we further demonstrate that vanilla gradient normalization can cause the iterates $\boldsymbol{x}_i^t$ to remain arbitrarily far from the optimal solution (see Appendix A for a proof).

**Claim 1.** *Consider algorithm 4. For any even $n$, for any $B \geq 1$, there exist $\{f_i\}_{i=1}^n$ satisfying Assumptions 1-2, a gradient oracle satisfying Assumption 3, a mixing matrix satisfying Assumption 4, and an initialization $\boldsymbol{x}_0$, such that the associated parameters $f_*, L, \sigma, \boldsymbol{W}, \boldsymbol{x}_0$, are independent of $B$. Then, $\forall T \geq 1, \forall \alpha > 0$, it holds that $\frac{1}{nT} \sum_{t=0}^{T-1} \sum_{i=1}^{n} \mathbb{E}\big[\|\nabla f(\boldsymbol{x}_i^t)\|\big] \geq B$.*

We next break the limitations of vanilla gradient normalization in Claim 1 by incorporating gradient tracking and momentum variance reduction. This enables the successful use of normalization to suppress heavy-tailed noise while maintaining optimal convergence despite the added nonlinearity.

## 4 MAIN RESULTS

We present the main convergence results of `GT-NSGDm` and discuss their implications. The detailed analyses are deferred to Appendix B. We first consider the case where the tail index $p$ is known.

**Theorem 1.** *Let Assumptions 1, 2, 3, 4 hold. Denote $f(\bar{\boldsymbol{x}}^0) - f_* = \Delta_0$, $[\nabla f_1(\bar{\boldsymbol{x}}^0), \ldots, \nabla f_n(\bar{\boldsymbol{x}}^0)]^\top = \nabla F(\mathbf{1}_n \otimes \bar{\boldsymbol{x}}^0)$. Take*

$$\alpha = \min\left(1, \sqrt{\frac{\Delta_0(1-\beta)(1-\lambda)}{4LT}}, \sqrt{\frac{\Delta_0(1-\lambda)}{3.5LT}}, \sqrt{\frac{(1-\lambda)^2\Delta_0}{2n^{\frac{1}{2}}LT}}\right), \quad (5)$$

*and $1 - \beta = 1/T^{\frac{p}{3p-2}}$. Assume $\beta \geq 1/10$, then the sequence generated by `GT-NSGDm` satisfies that*

$$\frac{1}{nT}\sum_{t=0}^{T-1}\sum_{i=1}^{n}\mathbb{E}\left[\|\nabla f(\boldsymbol{x}_i^t)\|\right] = O\left(\frac{\sigma}{n^{1-\frac{1}{p}}T^{\frac{p-1}{3p-2}}} + \frac{1}{T^{\frac{p-1}{3p-2}}}\sqrt{\frac{L\Delta_0}{1-\lambda}} + \frac{\|\nabla f(\bar{\boldsymbol{x}}^0)\|}{T^{\frac{2p-2}{3p-2}}} + \sqrt{\frac{3.5L\Delta_0}{(1-\lambda)T}}\right.$$

$$\left. + \sqrt{\frac{n^{\frac{1}{2}}L\Delta_0}{(1-\lambda)^2T}} + \frac{\sigma n^{\frac{1}{2}}}{(1-\lambda)^{\frac{1}{p}}T^{\frac{p}{3p-2}}} + \frac{\|\nabla F(\mathbf{1}_n \otimes \bar{\boldsymbol{x}}^0)\|}{(1-\lambda)n^{\frac{1}{2}}T^{\frac{p}{3p-2}}} + \frac{\sigma}{1-\lambda}\frac{n^{\frac{1}{2}}}{T^{\frac{2p-1}{3p-2}}} + \frac{\Delta_0}{T}\right).$$

*Remark* 4 (Order-optimal rate). Theorem 1 establishes a non-asymptotic upper bound on the mean $\ell_2$ norm stationary gap of `GT-NSGDm` over any finite time horizon $T$. The $O(\cdot)$ here only absorbs universal constants and preserves all problem parameters. It achieves the *optimal* $O\left(1/T^{\frac{p-1}{3p-2}}\right)$ convergence rate in terms of $T$ as it matches the lower bound proved in Zhang et al. (2020). This optimal guarantee is achieved in decentralized settings *for the first time.*

*Remark* 5 (Speedup in $n$). We discuss the asymptotic speedup in number of nodes $n$. For sufficiently large $T$ (or sufficiently small target optimality gap), the upper bound in Theorem 1 is dominated by the leading terms $(1/T^{\frac{p-1}{3p-2}})(\sigma/n^{1-1/p} + \sqrt{L\Delta_0/(1-\lambda)})$. In the high-noise regime $\sigma \gg n^{1-1/p}\sqrt{L\Delta_0/(1-\lambda)}$, the upper bound has a speedup factor $n^{1-1/p}$. In practice, the noise scale (measured by $\sigma$) in training attention models or in other high-dimensional problems can be very large, and the speedup in $n$ contributes as a noise reduction.

When the tail index $p$ is unknown in advance, we establish the following convergence rate.

**Theorem 2.** *Let Assumptions 1, 2, 3, 4 hold and take $\alpha$ as in (5). Take $1 - \beta = 1/\sqrt{T}$ and assume $\beta \geq 1/10$. Then `GT-NSGDm` guarantees that*

$$\frac{1}{nT}\sum_{t=0}^{T-1}\sum_{i=1}^{n}\mathbb{E}\left[\|\nabla f(\boldsymbol{x}_i^t)\|\right] \leq O\left(\frac{\sigma}{n^{1-\frac{1}{p}}T^{\frac{p-1}{2p}}} + \frac{1}{T^{\frac{1}{4}}}\sqrt{\frac{L\Delta_0}{1-\lambda}} + \frac{\|\nabla f(\bar{\boldsymbol{x}}^0)\|}{\sqrt{T}} + \sqrt{\frac{3.5L\Delta_0}{(1-\lambda)T}}\right.$$

$$\left. + \frac{\sigma n^{\frac{1}{2}}}{(1-\lambda)^{\frac{1}{p}}\sqrt{T}} + \frac{\|\nabla F(\mathbf{1}_n \otimes \bar{\boldsymbol{x}}^0)\|}{(1-\lambda)n^{\frac{1}{2}}\sqrt{T}} + \sqrt{\frac{n^{\frac{1}{2}}L\Delta_0}{(1-\lambda)^2T}} + \frac{\sigma n^{\frac{1}{2}}}{(1-\lambda)T^{\frac{2p-1}{2p}}} + \frac{\Delta_0}{T}\right).$$

Theorem 2 establishes an upper bound of $O(1/T^{\frac{p-1}{2p}})$ when the tail index $p$ is unknown, matching the best-known rate in the *centralized* setting where algorithm parameters do not rely on $p$ (Liu & Zhou, 2025). While the convergence rate in Yu et al. (2023) is also independent of the knowledge of $p$, it is only for strongly convex functions and its exact rate exponent remains unspecified.

*Remark* 6 (Speedup in $n$ and topology independent rate). Consider $p \in (1, 2)$, i.e., the heavy-tailed case with unbounded variance this paper focuses on. When $T$ is sufficiently large (as required to achieve sufficiently small target optimality gap), the upper bound in Theorem 2 is dominated by $\frac{\sigma}{n^{1-1/p}} \cdot \frac{1}{T^{(p-1)/2p}}$. Significantly, this upper bound is *independent* of network topology ($\lambda$) and exhibits a speedup factor $n^{1-1/p}$ in all regimes.

*Remark* 7 (Hyperparameter selection). We compare with the centralized work Liu & Zhou (2025) in terms of hyperparameter selections. When $p$ is known, the centralized setup requires $\Delta_0, L, \sigma, T$, while our decentralized setup requires $\Delta_0, L, T, \lambda, n$. Notably, our selection is independent of $\sigma$ but requires the network parameters $n, \lambda$. When $p$ is unknown, the centralized setup requires $L, T$, while the decentralized case additionally requires $\Delta_0, \lambda, n$. We note that $n$ is generally easy to estimate, and we give an example where our hyperparameter dependence on $\lambda$ can be eliminated. If all nodes

know $n$ and set weights $\boldsymbol{W} = \boldsymbol{I}_n - (1/n)\boldsymbol{L}$ where $\boldsymbol{L}$ is the graph Laplacian, one can obtain the estimate $\lambda \leq 1 - c/n^3$ (Mohar et al., 1991) for some universal constant $c$. Once we have such an upper bound, we can replace the terms involving $1/(1-\lambda)$ with this estimate in equation (35); then in the final proof steps, we can select a step size that is independent of $\lambda$ while still obtaining the same order-optimal rate.

*Remark* 8 (Dominant terms for small $T$). Both Theorems 1 and 2 order the upper bound terms in an increasing order of the rate exponent of $1/T$, so the dominant terms are clear for sufficiently large $T$. We briefly discuss some cases to illustrate the effects of other parameters including $n$, $\lambda$, and $p$ when $T$ is small. We discuss one varying parameter while assuming other parameters are fixed. First, for network connectivity $1-\lambda$, $1/T^{\frac{p-1}{3p-2}}$ has coefficient $\frac{1}{\sqrt{1-\lambda}}$ while the faster decaying term $\frac{1}{\sqrt{T}}$ has coefficient $\frac{1}{1-\lambda}$. When $1-\lambda \leq \frac{n^{1/2}}{T^{p/(3p-2)}}$, the latter term $\frac{1}{\sqrt{T}}\sqrt{\frac{n^{1/2}L\Delta_0}{(1-\lambda)^2}}$ will dominate. Second, for $n$, if $n^{1/2} \geq (1-\lambda)T^{\frac{p}{3p-2}}$, we will also have a dominant term $\frac{1}{\sqrt{T}}\sqrt{\frac{n^{1/2}L\Delta_0}{(1-\lambda)^2}}$. Third, for $p \in (1,2]$: When $p$ approaches 1 from above, $\frac{1}{T^{(p-1)/(3p-2)}}$ will have coefficient $\sqrt{\frac{L\Delta_0}{1-\lambda}} + \sigma$ and the speedup in terms of the number of nodes will vanish, but the change in $\frac{1}{T^{p/(3p-2)}}\frac{\sigma n^{1/2}}{(1-\lambda)^{1/p}}$ will not introduce a new dominant term if we focus on the dependence on $1-\lambda$ since there already exists a slower term $\frac{1}{\sqrt{T}}\frac{n^{1/2}}{1-\lambda}$. Similarly, when $p$ approaches 2, the speedup in $n$ is more significant; the term $\frac{1}{T^{p/(3p-2)}}\frac{\sigma n^{1/2}}{(1-\lambda)^{1/p}}$ only gets smaller and does not change the dominant terms. Although our rates are established for general $p \in (1,2]$, we can compare parameter dependence with rates obtained for $p = 2$ only. For example, in Xin et al. (2021), for large enough $T$, the upper bound is

$$O\left(\frac{\Delta_0 + \sigma\sqrt{L}}{n^{1/4}T^{1/4}} + \frac{\sqrt{n}\sigma L}{(1-\lambda^2)^{3/2}\sqrt{T}} + \frac{L\|\nabla F(\mathbf{1}_n \otimes \bar{\boldsymbol{x}}^0)\|}{(1-\lambda^2)^{3/2}T}\right).$$

If we set $p = 2$ in our bounds, dropping some higher order terms, we have

$$O\left(\frac{1}{T^{1/4}} \cdot \left(\sqrt{\frac{L\Delta_0}{1-\lambda}} + \frac{\sigma}{n^{1/2}}\right) + \frac{1}{\sqrt{T}} \cdot \left(\|\nabla f(\bar{\boldsymbol{x}}^0)\| + \sqrt{\frac{3.5L\Delta_0}{1-\lambda}} + \sqrt{\frac{n^{\frac{1}{2}}L\Delta_0}{(1-\lambda)^2}} + \frac{\sigma n^{\frac{1}{2}}}{(1-\lambda)^{1/2}}\right)\right).$$

We compare the above two bounds. For coefficients of $1/T^{1/4}$, our bound has dependence $1/n^{1/2}$ which is faster than the $1/n^{1/4}$ in the $p = 2$ case, but the dependence in $1/\sqrt{1-\lambda}$ is worse. For coefficients of $1/\sqrt{T}$, our bounds have dependence $1/(1-\lambda)$ while the $p = 2$ case has dependence $1/(1-\lambda^2)^{3/2}$, which is strictly larger than ours.

## 5 EXPERIMENTS

We assess the performance of GT-NSGDm through numerical experiments. We first conduct studies on synthetic datasets that mimic language modeling under controlled heavy-tailed noise injection, following Lee et al. (2025). We also present experiments on decentralized training of a decoder-only Transformer (GPT) model with 3M parameters on the Multi30k dataset.

**Baselines.** We compare GT-NSGDm with four decentralized baselines: DSGD(Nedic & Ozdaglar, 2009), GT-DSGD (Xin et al., 2020b), DSGD-Clip (Sun & Chen, 2024), and SClip-EF -Network (Yu et al., 2023). DSGD and GT-DSGD handle regular stochastic noise with bounded variance. DSGD-Clip converges for strongly convex functions under bounded domains or gradients (Sun & Chen, 2024). SClip-EF-Network achieves convergence under symmetric noise with bounded $\mathbb{E}[\|\boldsymbol{\xi}_i^t\|^p \mid \mathcal{F}_{t-1}]$ for $p = 1$. All methods are initialized identically and tuned via grid search. Detailed baseline descriptions appear in Table 2 (Appendix C.1).

**Graph topology.** We consider three graph topologies: undirected ring, directed exponential, and complete graphs (see Lian et al. (2017); Nedić et al. (2018); Assran et al. (2019)). Weight matrices use Metropolis weights (Xiao et al., 2005). For synthetic experiments, we set the number of nodes to $n = 20$, we obtain $\lambda = 0.904, 0.714$, and $0$ for the ring, exponential, and complete graphs, respectively. For Transformer training with $n = 8$, we have corresponding $\lambda = 0.804, 0.6$, and $0$.

## 5.1 ROBUST LINEAR REGRESSION ON SYNTHETIC TOKENIZED DATA

We use this synthetic experiment to test our convergence rates under controlled heavy-tailed noise. We consider nonconvex regularized linear regression on synthetic data mimicking language tokens. In language modeling, token frequencies exhibit heavy-tailed distributions: few tokens appear frequently, while most are rare but contextually important. We construct the following synthetic dataset $X$ of 1k samples of dimension $d = 20$. The first four features simulate frequent tokens and are sampled independently from Bernoulli distributions: the first two from $\text{Bern}(0.9)$ and the next two from $\text{Bern}(0.5)$. The remaining 16 features represent rare tokens, each sampled from $\text{Bern}(0.1)$. The optimal weight $w_*$ is Gaussian-sampled, with labels $y = X w_*$. The synthetic dataset $(X, y)$ is evenly distributed over $n = 20$ nodes, where each node $i$ holds a sub-dataset $(X_i, y_i)$, estimate $w_i$, and a linear regression model with nonconvex robust Tukey's biweight loss function (Beaton & Tukey, 1974) to estimate $w_*$. We inject three different *zero-mean* noises, Gaussian noise ($\mathcal{N}(\mathbf{0}, 3I_d)$), Student's $t$ noise (degrees of freedom 1.5, scale 1.0), and Lévy $\alpha$-stable noise (stability parameter 1.5, skewness parameter 0.5, scale 1.0, non-symmetric, multiplied by 0.1) into the exact gradient, using corrupted stochastic gradients for updates. See Appendix C.2 for additional details.

In Figure 2, we evaluate `GT-NSGDm` against baselines on ring graphs under various gradient noise. `DSGD` and `GT-DSGD` converge under Gaussian noise but become unstable under heavy-tailed noise. `DSGD-Clip` remains stable but fails to reach the optimum. Both `GT-NSGDm` and `SClip-EF-Network` exhibit robust convergence and near-optimal performance across all scenarios, consistent with their theoretical guarantees under heavy-tailed noise.

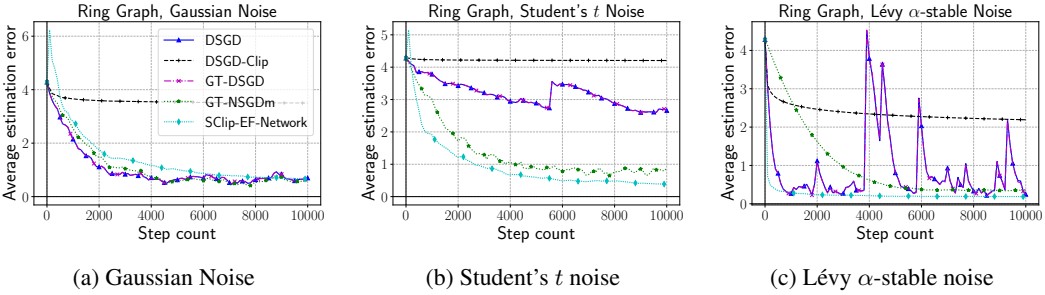

(a) Gaussian Noise      (b) Student's $t$ noise      (c) Lévy $\alpha$-stable noise

Figure 2: Comparison of performance on a ring graph under various types of injected stochastic gradient noise, measured by the average estimation error $(1/n) \sum_{i=1}^{n} \|w_i^t - w_*\|$ over step count $t$.

In Figure 3, we test `GT-NSGDm`'s dependence on connectivity ($\lambda$), noise level ($\sigma$), and the number of nodes ($n$), varying each while fixing others. In Figure 3(a), we inject Lévy $\alpha$-stable noise and test the performance of `GT-NSGDm` on ring, directed exponential, undirected exponential, and complete graphs with $\lambda = 0.904, 0.714, 0.6, 0$, respectively. `GT-NSGDm` achieves comparable final errors under weak connectivity (i.e., large $\lambda$) versus complete graphs, showing favorable dependence on network connectivity under heavy-tailed noise. In Figure 3(b), we evaluate `GT-NSGDm`'s performance under different noise levels on a directed exponential graph. Under Gaussian noise with scale 1 (unit variance), `GT-NSGDm` reaches the best optimality; the final error increases as $\sigma$ grows, as observed under both Gaussian and Lévy $\alpha$-stable noise. In Figure 3(c), we inject Lévy $\alpha$-stable noise on complete graphs ($\lambda = 0$ for all $n$) with varying number of nodes. As $n$ increases from 2 to 40, convergence speed improves with final errors $[0.4, 0.35, 0.29, 0.20, 0.21]$, demonstrating speedup over certain $n$ ranges, supporting theoretical discussions in Remarks 5 and 6.

## 5.2 DECENTRALIZED TRAINING OF TRANSFORMERS

We evaluate `GT-NSGDm`'s empirical performance on language modeling using a 3M-parameter GPT model (Radford et al., 2018) for auto-regressive modeling on Multi30k (29k sentences, 4.4M tokens). We assess performance using validation log-perplexity. On 8-node graphs with three topologies, we distribute training data evenly and initialize identical GPT models per node. We introduce three additional baselines: `DSGD-GClip` (DSGD with constant step size and $\ell_2$ gradient clipping level), `DSGD-CClip` (DSGD with constant step size and component-wise gradient clipping level), `QG-DSGDm` (Lin et al., 2021) and `GT-Adam` (Carnevale et al., 2022) (all without theoretical guar-

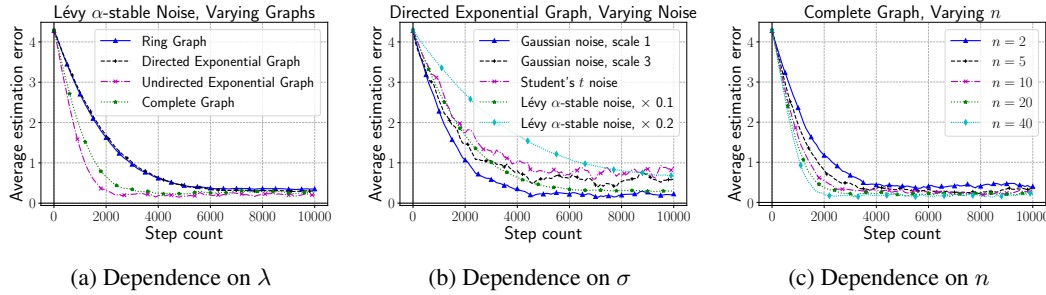

(a) Dependence on $\lambda$        (b) Dependence on $\sigma$        (c) Dependence on $n$

Figure 3: Empirical studies on `GT-NSGDm`'s dependence on problem parameters $\lambda, \sigma, n$.

antees under heavy-tailed noise; see Table 2 in Appendix C.1). We run all methods for 12 epochs with batch size 64. See Appendix C.3 for model and hyperparameter details.

Table 1 presents average validation loss and standard deviation over five independent runs for each algorithm across three topologies. Results show that `GT-NSGDm` nearly matches the best baseline `DSGD-GClip` (which lacks theoretical guarantees under heavy-tailed gradient noise) while significantly outperforming the other two theoretically-guaranteed baselines across all topologies. We note that this decentralized training experiment is simulated to demonstrate algorithm effectiveness and has practical limitations.

Table 1: Topologies ring, directed exponential (Exp.), and complete (Comp.) graphs. Algorithms are grouped by theoretical (Theo.) guarantees under heavy-tailed noise: with (w/) or without (w/o).

| Algorithms | Theo. | Ring | Exp. | Comp. |
|---|---|---|---|---|
| `DSGD` | w/o | $5.633_{\pm 0.008}$ | $5.632_{\pm 0.007}$ | $5.635_{\pm 0.007}$ |
| `DSGD-GClip` | w/o | $\mathbf{0.253}_{\pm 0.007}$ | $\mathbf{0.249}_{\pm 0.010}$ | $\mathbf{0.267}_{\pm 0.010}$ |
| `DSGD-CClip` | w/o | $2.725_{\pm 3.179}$ | $5.058_{\pm 2.388}$ | $8.225_{\pm 1.695}$ |
| `GT-DSGD` | w/o | $5.362_{\pm 0.002}$ | $5.632_{\pm 0.002}$ | $5.631_{\pm 0.002}$ |
| `GT-Adam` | w/o | $0.520_{\pm 0.038}$ | $0.587_{\pm 0.096}$ | $0.524_{\pm 0.045}$ |
| `QG-DSGDm` | w/o | $0.394_{\pm 0.007}$ | $0.388_{\pm 0.013}$ | $0.353_{\pm 0.011}$ |
| `SClip-EF-Network` | w/ | $5.653_{\pm 0.012}$ | $5.632_{\pm 0.003}$ | $5.636_{\pm 0.004}$ |
| `DSGD-Clip` | w/ | $5.633_{\pm 0.004}$ | $5.659_{\pm 0.013}$ | $5.661_{\pm 0.006}$ |
| `GT-NSGDm` | w/ | $\mathbf{0.258}_{\pm 0.007}$ | $\mathbf{0.261}_{\pm 0.007}$ | $\mathbf{0.282}_{\pm 0.009}$ |

## 6 CONCLUSION AND FUTURE WORK

In this paper, we have proposed `GT-NSGDm` for solving decentralized nonconvex smooth optimization to address heavy-tailed noise. The key idea is to leverage normalization, together with momentum variance reduction, to combat heavy-tailed noise, and use gradient tracking to handle cross-node heterogeneity and the nonlinearity introduced by normalization. Theoretical analyses establish that `GT-NSGDm` attains an optimal convergence rate when the tail index $p$ is known, and a rate that matches the best centralized one when $p$ is unknown. Extensive experiments on nonconvex linear regression and decentralized Transformer training show that `GT-NSGDm` is robust and efficient under heavy-tailed noise across various topologies, and achieves a speedup in $n$. Future directions include extending the current analysis to other nonlinearities, such as sign and clipping (Zhang et al., 2020), and generalizing `GT-NSGDm` to handle objective functions under relaxed smoothness conditions (Liu & Zhou, 2025).

## ACKNOWLEDGMENTS

We thank the anonymous reviewers for their valuable feedback that improved the presentation of this paper and for identifying minor errors in the proofs. The work of S. Yu and S. Kar is supported in part by NSF grant ECCS 2330196. The work of D. Jakovetic was supported by the Ministry of Science, Technological Development, and Innovation (Grants No. 451-03-33/2026-03/ 200125 & 451-03-34/2026-03/ 200125); and the Science Fund of Republic of Serbia, project "LASCADO" (grant 7359).

## REPRODUCIBLE STATEMENT

We provide detailed proofs for Claim 1 in Appendix A and for our main theoretical results, Theorems 1 and 2, in Appendix B. In Appendix C, we provide detailed hardware configurations, algorithm descriptions, and hyperparameter settings for our numerical experiments.

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
