APPENDIX

## A    PROOF OF CLAIM 1

*Proof.* Consider $n$ scalar functions that for each $i \in \mathcal{V}$, $f_i(x) = (1/2)(x - a_i)^2$ for some $a_i$, and complete graph with $\boldsymbol{W} = (1/n)\mathbf{1}_n\mathbf{1}_n^\top$. Let $a_i = a, \forall i = 1, \ldots, n/2$, and $a_i = b, \forall i = n/2+1, \ldots, n$, and $b - a > 2B+1$. Let $x_i^0 = a + 0.5, \forall i \in \mathcal{V}$. Then, $\forall i \in \mathcal{V}$, vanilla normalization reduces to

$$
\begin{aligned}
x_i^1 &= \frac{1}{n} \sum_{r=1}^n \left( x_r^0 - \alpha \operatorname{sign}(x_r^0 - a_r) \right) \\
&= x_r^0 - \frac{\alpha}{n} \sum_{r=1}^n \operatorname{sign}(x_r^0 - a_r) \\
&= x_r^0 - \frac{\alpha}{n} \sum_{r=1}^{n/2} \operatorname{sign}(0.5) - \frac{\alpha}{n} \sum_{r=n/2+1}^n \operatorname{sign}(0.5 - (b-a)) \\
&= x_r^0.
\end{aligned}
$$

Therefore, $x_r^t = a + 0.5, \forall r \in \mathcal{V}, \forall t \geq 0$. Since the optimal solution to the original problem is $\frac{a+b}{2}$, the optimality gap is $\frac{b-a}{2} - 0.5 \geq B$. $\qquad\square$

*Remark* 9. Note that the proof above can be further extended to the case where the gradient oracle admits almost surely bounded gradient noise. We can use the noise bound to adapt the choices of $a, b, \varepsilon$ such that all signs still get canceled. Similar examples have been used to show divergence results in Shulgin et al. (2025).

## B    PROOFS OF THEOREMS

**Proof structure**. The central recursion of our analysis leverages the descent lemma for $L$ smooth functions applied to consecutive network averages of $\{x_i^{t+1}\}$ and $\{x_i^t\}$ (Lemma 3). This recursion involves two coupled error sources: consensus errors between $\{x_i^t\}, \{y_i^t\}$, and gradient estimation errors. We establish the intricate coupling between these errors through a series of intermediate lemmas. Our proof strategy proceeds as follows: we first derive two key lemmas that bound consensus errors in terms of gradient estimation errors (Lemma 4 and 6), then decompose gradient estimation errors into constituent components, including average gradient estimation errors (Lemma 7) and stack gradient estimation errors (Lemma 8), and bound each separately. With all error bounds established, we substitute these into the main recursion and optimize hyperparameter selection to achieve the final order-optimal convergence rates.

### B.1    PRELIMINARIES

We define some stacked long vectors,

$$
\begin{aligned}
F(\boldsymbol{x}^t) &:= [f_1(\boldsymbol{x}_1^t), \ldots, f_n(\boldsymbol{x}_n^t)]^\top, \\
\nabla F(\boldsymbol{x}^t) &:= [\nabla f_1(\boldsymbol{x}_1^t)^\top, \ldots, \nabla f_n(\boldsymbol{x}_n^t)^\top]^\top, \\
\boldsymbol{g}(\boldsymbol{x}^t, \boldsymbol{\xi}^t) &:= [\boldsymbol{g}_1(\boldsymbol{x}_1^t, \boldsymbol{\xi}_1^t)^\top, \ldots, \boldsymbol{g}_n(\boldsymbol{x}_n^t, \boldsymbol{\xi}_n^t)^\top]^\top \\
\boldsymbol{v}^t &:= [(\boldsymbol{v}_1^t)^\top, \ldots, (\boldsymbol{v}_n^t)^\top]^\top, \\
\mathcal{N}(\boldsymbol{y}^t) &:= \left[ \frac{(\boldsymbol{y}_1^t)^\top}{\|\boldsymbol{y}_1^t\|}, \ldots, \frac{(\boldsymbol{y}_n^t)^\top}{\|\boldsymbol{y}_n^t\|} \right]^\top, \\
\boldsymbol{x}^t &:= [(\boldsymbol{x}_1^t)^\top, \ldots, (\boldsymbol{x}_n^t)^\top]^\top.
\end{aligned}
$$

Then, Algorithm 1 can be rewritten in the compact long-vector form:

$$\boldsymbol{v}^t = \beta \boldsymbol{v}^{t-1} + (1-\beta)\boldsymbol{g}(\boldsymbol{x}^t, \boldsymbol{\xi}^t); \tag{6}$$

$$\boldsymbol{y}^t = (\boldsymbol{W} \otimes \boldsymbol{I}_d)(\boldsymbol{y}^{t-1} + \boldsymbol{v}^t - \boldsymbol{v}^{t-1}), \tag{7}$$

$$\boldsymbol{x}^{t+1} = (\boldsymbol{W} \otimes \boldsymbol{I}_d)(\boldsymbol{x}^t - \alpha \mathcal{N}(\boldsymbol{y}^t)). \tag{8}$$

We define the following averages over network:

$$\bar{\boldsymbol{v}}^t = \frac{1}{n}\sum_{i=1}^n \boldsymbol{v}_i^t, \quad \bar{\boldsymbol{y}}^t = \frac{1}{n}\sum_{i=1}^n \boldsymbol{y}_i^t, \quad \tilde{\boldsymbol{y}}^t = \frac{1}{n}\sum_{i=1}^n \frac{\boldsymbol{y}_i^t}{\|\boldsymbol{y}_i^t\|}, \quad \bar{\boldsymbol{x}}^t = \frac{1}{n}\sum_{i=1}^n \boldsymbol{x}_i^t, \quad \overline{\nabla}F(\boldsymbol{x}^t) = \frac{1}{n}\sum_{i=1}^n \nabla f_i(\boldsymbol{x}_i^t). \tag{9}$$

From the doubly stochasticity of $\boldsymbol{W}$, the global average updates as

$$\bar{\boldsymbol{x}}^{t+1} = \bar{\boldsymbol{x}}^t - \frac{\alpha}{n}\sum_{r=1}^n \frac{\boldsymbol{y}_r^t}{\|\boldsymbol{y}_r^t\|} = \bar{\boldsymbol{x}}^t - \alpha\tilde{\boldsymbol{y}}^t. \tag{10}$$

## B.2 INTERMEDIATE LEMMAS

We first present some standard useful relations to be used in our analysis.

**Lemma 1.** *The following relations hold:*

1. $\bar{\boldsymbol{y}}^t = \bar{\boldsymbol{v}}^t$;

2. $\boldsymbol{W} - \boldsymbol{1}_n\boldsymbol{1}_n^\top/n = (\boldsymbol{W} - \boldsymbol{1}_n\boldsymbol{1}_n^\top/n)(\boldsymbol{I}_n - \boldsymbol{1}_n\boldsymbol{1}_n^\top/n) = (\boldsymbol{I}_n - \boldsymbol{1}_n\boldsymbol{1}_n^\top/n)(\boldsymbol{W} - \boldsymbol{1}_n\boldsymbol{1}_n^\top/n)$;

3. $\boldsymbol{W}^k - \boldsymbol{1}_n\boldsymbol{1}_n^\top/n = (\boldsymbol{W} - \boldsymbol{1}_n\boldsymbol{1}_n^\top/n)^k, \forall k \in \mathbb{N}_+$;

4. $(1/\sqrt{n})\sum_{i=1}^n \|\boldsymbol{a}_i\| \le \|\boldsymbol{a}\| \le \sum_{i=1}^n \|\boldsymbol{a}_i\|, \forall \boldsymbol{a} = [\boldsymbol{a}_1^\top, \ldots, \boldsymbol{a}_n^\top]^\top \in \mathbb{R}^{nd}$,

5. $\sum_{i=1}^m a_i^p \le \left(\sum_{i=1}^m a_i\right)^p \le m^{p-1}\sum_{i=1}^m a_i^p, \forall m \in \mathbb{N}_+, \forall a_i \in \mathbb{R}_+$.

We then present a standard decent lemma for $L$-smooth functions .

**Lemma 2** (Decent lemma for $L$-smooth functions)**.** *Let Assumption 2 hold. For any $\boldsymbol{x}, \boldsymbol{y} \in \mathbb{R}^d$, there holds*

$$f(\boldsymbol{y}) \le f(\boldsymbol{x}) + \nabla f(\boldsymbol{x})^\top(\boldsymbol{y} - \boldsymbol{x}) + \frac{L}{2}\|\boldsymbol{x} - \boldsymbol{y}\|^2.$$

We next present the main descent lemma on the network average.

**Lemma 3** (Decent lemma for network average)**.** *Let Assumption 2 hold. Let $\epsilon^t = \bar{\boldsymbol{y}}^t - \nabla f(\bar{\boldsymbol{x}}^t)$. We have*

$$\sum_{t=0}^{T-1} \alpha\|\nabla f(\bar{\boldsymbol{x}}^t)\| \le f(\bar{\boldsymbol{x}}^0) - f_* + \sum_{t=0}^{T-1} 2\alpha\|\epsilon^t\| + \sum_{t=0}^{T-1} \frac{\alpha}{n}\sum_{i=1}^n \|\bar{\boldsymbol{y}}^t - \boldsymbol{y}_i^t\| + \sum_{t=0}^{T-1} \frac{L}{2}\alpha^2.$$

*Proof.* Since $\|\bar{\boldsymbol{x}}^{t+1} - \bar{\boldsymbol{x}}^t\| = \alpha\|\tilde{\boldsymbol{y}}^t\| = \alpha$, applying Lemma 2 on $\bar{\boldsymbol{x}}^{t+1}, \bar{\boldsymbol{x}}^t$ gives that

$$f(\bar{\boldsymbol{x}}^{t+1}) \le f(\bar{\boldsymbol{x}}^t) + \nabla f(\bar{\boldsymbol{x}}^t)^\top(\bar{\boldsymbol{x}}^{t+1} - \bar{\boldsymbol{x}}^t) + \frac{L}{2}\|\bar{\boldsymbol{x}}^{t+1} - \bar{\boldsymbol{x}}^t\|^2$$

$$\overset{(i)}{\le} f(\bar{\boldsymbol{x}}^t) - \alpha(\bar{\boldsymbol{y}}^t - \epsilon^t)^\top\tilde{\boldsymbol{y}}^t + \frac{L}{2}\alpha^2$$

$$\overset{(ii)}{\le} f(\bar{\boldsymbol{x}}^t) - \alpha(\bar{\boldsymbol{y}}^t)^\top\tilde{\boldsymbol{y}}^t + \alpha\|\epsilon^t\| + \frac{L}{2}\alpha^2, \tag{11}$$

where we used the definitions (9)(10) in $(i)$, and used Cauchy-Schwartz inequality followed by $\|\tilde{\boldsymbol{y}}^t\| \leq 1$ in $(ii)$. Next,

$$
\begin{aligned}
-(\bar{\boldsymbol{y}}^t)^\top \tilde{\boldsymbol{y}}^t &= -(\bar{\boldsymbol{y}}^t)^\top \Big[ \frac{\bar{\boldsymbol{y}}^t}{\|\bar{\boldsymbol{y}}^t\|} + \frac{1}{n} \sum_{i=1}^n \boldsymbol{y}_i^t \big( \frac{1}{\|\boldsymbol{y}_i^t\|} - \frac{1}{\|\bar{\boldsymbol{y}}^t\|} \big) \Big] \\
&\leq -\|\bar{\boldsymbol{y}}^t\| + \| \frac{1}{n} \sum_{i=1}^n \boldsymbol{y}_i^t \big( \frac{\|\bar{\boldsymbol{y}}^t\|}{\|\boldsymbol{y}_i^t\|} - 1 \big) \| \\
&\overset{(i)}{\leq} -\|\nabla f(\bar{\boldsymbol{x}}^t)\| + \|\boldsymbol{\epsilon}^t\| + \frac{1}{n} \sum_{i=1}^n \big| \|\bar{\boldsymbol{y}}^t\| - \|\boldsymbol{y}_i^t\| \big| \\
&\overset{(ii)}{\leq} -\|\nabla f(\bar{\boldsymbol{x}}^t)\| + \|\boldsymbol{\epsilon}^t\| + \frac{1}{n} \sum_{i=1}^n \|\bar{\boldsymbol{y}}^t - \boldsymbol{y}_i^t\|,
\end{aligned}
\tag{12}
$$

where we used $\|\bar{\boldsymbol{y}}^t\| = \|\nabla f(\bar{\boldsymbol{x}}^t) + \boldsymbol{\epsilon}^t\| \geq \|\nabla f(\bar{\boldsymbol{x}}^t)\| - \|\boldsymbol{\epsilon}^t\|$, and Cauchy-Schwartz inequality in $(i)$, and $\big| \|\boldsymbol{a}\| - \|\boldsymbol{b}\| \big| \leq \|\boldsymbol{a} - \boldsymbol{b}\|$ for any $\boldsymbol{a}, \boldsymbol{b} \in \mathbb{R}^d$ in $(ii)$. Plugging in (12) into (11), and summing over $t = 0, \ldots, T-1$, we have

$$
f(\bar{\boldsymbol{x}}^T) \leq f(\bar{\boldsymbol{x}}^0) - \sum_{t=0}^{T-1} \alpha \|\nabla f(\bar{\boldsymbol{x}}^t)\| + \sum_{t=0}^{T-1} 2\alpha \|\boldsymbol{\epsilon}^t\| + \sum_{t=0}^{T-1} \frac{\alpha}{n} \sum_{i=1}^n \|\bar{\boldsymbol{y}}^t - \boldsymbol{y}_i^t\| + \sum_{t=0}^{T-1} \frac{L}{2} \alpha^2.
$$

Using $f(\bar{\boldsymbol{x}}^T) \geq f_*$ and rearranging terms above give the desired result. □

With Lemma 3, it remains to bound the gradient estimation error $\|\boldsymbol{\epsilon}^t\|$ and the consensus error $\boldsymbol{y}_i^t - \bar{\boldsymbol{y}}^t$. Let us decompose the gradient estimation error as follows:

$$
\boldsymbol{\epsilon}^t = \bar{\boldsymbol{y}}^t - \nabla f(\bar{\boldsymbol{x}}^t) = \bar{\boldsymbol{v}}^t - \nabla f(\bar{\boldsymbol{x}}^t) = \underbrace{\bar{\boldsymbol{v}}^t - \overline{\nabla} F(\boldsymbol{x}^t)}_{:=\boldsymbol{\epsilon}_1^t \in \mathbb{R}^d} + \underbrace{\overline{\nabla} F(\boldsymbol{x}^t) - \nabla f(\bar{\boldsymbol{x}}^t)}_{:=\boldsymbol{\epsilon}_2^t \in \mathbb{R}^d}.
\tag{13}
$$

It is clear that $\boldsymbol{\epsilon}_1^t$ is the gradient estimation error, and $\boldsymbol{\epsilon}_2^t$, exploiting the smoothness property in 2, can be bounded by the consensus error $\boldsymbol{x}_i^t - \bar{\boldsymbol{x}}^t$. Since the consensus error is also used in bounding $\boldsymbol{\epsilon}_1^t$, we need to first bound the consensus errors $\boldsymbol{x}_i^t - \bar{\boldsymbol{x}}^t$ and $\boldsymbol{y}_i^t - \bar{\boldsymbol{y}}^t$.

**Lemma 4** (Consensus errors of $\{\boldsymbol{x}_i^t\}$). *We have for all $t = 0, \ldots, T$,*

$$
\frac{1}{n} \sum_{i=1}^n \|\boldsymbol{x}_i^t - \bar{\boldsymbol{x}}^t\| \leq \frac{\alpha \lambda}{1 - \lambda}.
\tag{14}
$$

*Proof.* Using the relation 4 in Lemma 1 we have

$$
\frac{1}{n} \sum_{i=1}^n \|\boldsymbol{x}_i^t - \bar{\boldsymbol{x}}^t\| \leq \frac{1}{\sqrt{n}} \|\boldsymbol{x}^t - \boldsymbol{1}_n \otimes \bar{\boldsymbol{x}}^t\|.
\tag{15}
$$

From the compact form update in (8), we have

$$
\boldsymbol{x}^t = \big( \boldsymbol{W} \otimes \boldsymbol{I}_d \big) \boldsymbol{x}^0 - \alpha \sum_{k=0}^{t-1} (\boldsymbol{W} \otimes \boldsymbol{I}_d)^{t-k} \mathcal{N}(\boldsymbol{y}^k).
$$

It follows that

$$\|\boldsymbol{x}^t - \mathbf{1}_n \otimes \bar{\boldsymbol{x}}^t\|$$

$$= \|\big(\boldsymbol{I}_{nd} - \frac{1}{n}\mathbf{1}_n\mathbf{1}_n^\top \otimes \boldsymbol{I}_d\big)\boldsymbol{x}^t\|$$

$$\stackrel{(i)}{=} \alpha\|\sum_{k=0}^{t-1}\big(\boldsymbol{I}_{nd} - \frac{1}{n}\mathbf{1}_n\mathbf{1}_n^\top \otimes \boldsymbol{I}_d\big)\big(\boldsymbol{W} \otimes \boldsymbol{I}_d\big)^{t-k}\mathcal{N}(\boldsymbol{y}^k)\|$$

$$\leq \alpha\|\sum_{k=0}^{t-1}\|\boldsymbol{W}^{t-k} - \frac{1}{n}\mathbf{1}_n\mathbf{1}_n^\top\|_2\|\mathcal{N}(\boldsymbol{y}^k)\|$$

$$\stackrel{(ii)}{\leq} \alpha\|\sum_{k=0}^{t-1}\|\boldsymbol{W} - \frac{1}{n}\mathbf{1}_n\mathbf{1}_n^\top\|_2^{t-k}\|\mathcal{N}(\boldsymbol{y}^k)\|$$

$$\leq \alpha\sqrt{n}\sum_{k=0}^{t-1}\lambda^{t-k} \tag{16}$$

$$\stackrel{(iii)}{\leq} \frac{\alpha\sqrt{n}\lambda}{1-\lambda}. \tag{17}$$

where we used the double stochasticity of $\boldsymbol{W}$ and $\boldsymbol{x}_i^0 = \bar{\boldsymbol{x}}^0, \forall i \in [n]$ in $(i)$, the relation 3 in Lemma 1 in $(ii)$, and Assumption 4 in $(iii)$. Substituting (17) into (15) gives the desired bound in (14). □

Before proceeding to bound consensus errors for $\{\boldsymbol{y}_i^t\}$, we present the following bound on vector-valued martingale difference sequence from Liu & Zhou (2025).

**Lemma 5.** *Given a sequence of random vectors $\boldsymbol{d}_t \in \mathbb{R}^d, \forall t$ such that $\mathbb{E}[\boldsymbol{d}_t \mid \mathcal{F}_{t-1}] = \mathbf{0}$ where $\mathcal{F}_t = \sigma(\boldsymbol{d}_1, \ldots, \boldsymbol{d}_t)$ is the natural filtration, then for any $p \in [1, 2]$, there is*

$$\mathbb{E}\Big[\|\sum_{t=1}^T \boldsymbol{d}_t\|\Big] \leq 2\sqrt{2}\mathbb{E}\Big[\big(\sum_{t=1}^T \|\boldsymbol{d}_t\|^p\big)^{\frac{1}{p}}\Big], \forall T \geq 0.$$

**Lemma 6** (Consensus errors for $\{\boldsymbol{y}_i^t\}$). *We have for all $t = 0, \ldots, T$,*

$$\frac{1}{n}\mathbb{E}\big[\sum_{i=1}^n \|\boldsymbol{y}_i^t - \bar{\boldsymbol{y}}^t\|\big]$$

$$\leq 2\sqrt{2}n^{\frac{1}{2}}\big(\frac{1}{\beta} - 1\big)\Big(\sum_{k=0}^t \lambda^{(t-k+1)p}\Big)^{\frac{1}{p}}\sigma + \frac{1}{\sqrt{n}}\big(\frac{1}{\beta} - 1\big)\sum_{k=0}^t \lambda^{t-k+1}\mathbb{E}\big[\|\nabla F(\boldsymbol{x}^k) - \boldsymbol{v}^k\|\big].$$

*Proof.* Similar to (15), we have

$$\frac{1}{n}\sum_{i=1}^n \|\boldsymbol{y}_i^t - \bar{\boldsymbol{y}}^t\| \leq \frac{1}{\sqrt{n}}\|\boldsymbol{y}^t - \mathbf{1}_n \otimes \bar{\boldsymbol{y}}^t\|. \tag{18}$$

Following from (7),

$$\boldsymbol{y}^t - \mathbf{1}_n \otimes \bar{\boldsymbol{y}}^t \tag{19}$$

$$\stackrel{(7)}{=} \big(\boldsymbol{I}_{nd} - \frac{1}{n}\mathbf{1}_n\mathbf{1}_n^\top \otimes \boldsymbol{I}_d\big)\big(\boldsymbol{W} \otimes \boldsymbol{I}_d\big)(\boldsymbol{y}^{t-1} + \boldsymbol{v}^t - \boldsymbol{v}^{t-1})$$

$$= \big(\boldsymbol{W} \otimes \boldsymbol{I}_d - \frac{1}{n}\mathbf{1}_n\mathbf{1}_n^\top \otimes \boldsymbol{I}_d\big)\boldsymbol{y}^{t-1} + \big(\boldsymbol{W} \otimes \boldsymbol{I}_d - \frac{1}{n}\mathbf{1}_n\mathbf{1}_n^\top \otimes \boldsymbol{I}_d\big)(\boldsymbol{v}^t - \boldsymbol{v}^{t-1})$$

$$\stackrel{(i)}{=} \big(\boldsymbol{W} \otimes \boldsymbol{I}_d - \frac{1}{n}\mathbf{1}_n\mathbf{1}_n^\top \otimes \boldsymbol{I}_d\big)\big(\boldsymbol{I}_{nd} - \frac{1}{n}\mathbf{1}_n\mathbf{1}_n^\top \otimes \boldsymbol{I}_d\big)\boldsymbol{y}^{t-1} + \big(\boldsymbol{W} \otimes \boldsymbol{I}_d - \frac{1}{n}\mathbf{1}_n\mathbf{1}_n^\top \otimes \boldsymbol{I}_d\big)(\boldsymbol{v}^t - \boldsymbol{v}^{t-1})$$

$$\stackrel{(ii)}{=} \sum_{k=0}^t \big(\boldsymbol{W} \otimes \boldsymbol{I}_d - \frac{1}{n}\mathbf{1}_n\mathbf{1}_n^\top \otimes \boldsymbol{I}_d\big)^{t-k+1}(\boldsymbol{v}^t - \boldsymbol{v}^{t-1}), \tag{20}$$

where we used relation 3 in Lemma 1 in $(i)$ and used $\boldsymbol{y}_i^0 = \boldsymbol{0}_d, \forall i \in [n]$ in $(ii)$. From the update in (6), we have

$$\boldsymbol{v}^t - \boldsymbol{v}^{t-1} = (\beta - 1)\boldsymbol{v}^{t-1} + (1-\beta)\boldsymbol{g}(\boldsymbol{x}^t, \boldsymbol{\xi}^t) = (1-\beta)(\boldsymbol{v}^t - \boldsymbol{v}^{t-1}) + (1-\beta)(\boldsymbol{g}(\boldsymbol{x}^t, \boldsymbol{\xi}^t) - \boldsymbol{v}^t).$$

Then, there holds,

$$\boldsymbol{v}^t - \boldsymbol{v}^{t-1} = (\frac{1}{\beta} - 1)(\boldsymbol{g}(\boldsymbol{x}^t, \boldsymbol{\xi}^t) - \boldsymbol{v}^t) = (\frac{1}{\beta} - 1)(\boldsymbol{g}(\boldsymbol{x}^t, \boldsymbol{\xi}^t) - \nabla F(\boldsymbol{x}^t) + \nabla F(\boldsymbol{x}^t) - \boldsymbol{v}^t). \quad (21)$$

Putting the relation above into (20) and applying (20) recursively, from $\boldsymbol{y}_i^0 = \bar{\boldsymbol{y}}^0$, we have

$$\|\boldsymbol{y}^t - \boldsymbol{1}_n \otimes \bar{\boldsymbol{y}}^t\| \leq (\frac{1}{\beta} - 1)\|\sum_{k=0}^t (\boldsymbol{W} \otimes \boldsymbol{I}_d - \frac{1}{n}\boldsymbol{1}_n\boldsymbol{1}_n^\top \otimes \boldsymbol{I}_d)^{t-k+1}(\boldsymbol{g}(\boldsymbol{x}^k, \boldsymbol{\xi}^k) - \nabla F(\boldsymbol{x}^k))\|$$

$$+ (\frac{1}{\beta} - 1)\|\sum_{k=0}^t (\boldsymbol{W} \otimes \boldsymbol{I}_d - \frac{1}{n}\boldsymbol{1}_n\boldsymbol{1}_n^\top \otimes \boldsymbol{I}_d)^{t-k+1}(\nabla F(\boldsymbol{x}^k) - \boldsymbol{v}^k)\|$$

$$(22)$$

We note that the first half of the right hand side above can be addressed by Lemma 5:

$$\mathbb{E}\Big[\|\sum_{k=0}^t (\boldsymbol{W} \otimes \boldsymbol{I}_d - \frac{1}{n}\boldsymbol{1}_n\boldsymbol{1}_n^\top \otimes \boldsymbol{I}_d)^{t-k+1}(\boldsymbol{g}(\boldsymbol{x}^k, \boldsymbol{\xi}^k) - \nabla F(\boldsymbol{x}^k))\|\Big]$$

$$(23)$$

$$\leq 2\sqrt{2}\mathbb{E}\Big[\Big(\sum_{k=0}^t \lambda^{(t-k+1)p}\|\boldsymbol{g}(\boldsymbol{x}^k, \boldsymbol{\xi}^k) - \nabla F(\boldsymbol{x}^k)\|^p\Big)^{\frac{1}{p}}\Big].$$

We observe that

$$2\sqrt{2}\mathbb{E}\Big[\Big(\sum_{k=0}^t \lambda^{(t-k+1)p}\|\boldsymbol{g}(\boldsymbol{x}^k, \boldsymbol{\xi}^k) - \nabla F(\boldsymbol{x}^k)\|^p\Big)^{\frac{1}{p}} \mid \mathcal{F}_{t-1}\Big]$$

$$\leq 2\sqrt{2}\mathbb{E}\Big[\Big(\sum_{k=0}^t \lambda^{(t-k+1)p}\big(\sum_{i=1}^n \|\boldsymbol{g}_i(\boldsymbol{x}_i^k, \boldsymbol{\xi}_i^k) - \nabla f_i(\boldsymbol{x}_i^k)\|\big)^p\Big)^{\frac{1}{p}} \mid \mathcal{F}_{t-1}\Big]$$

$$\overset{(i)}{\leq} 2\sqrt{2}\mathbb{E}\Big[\Big(\sum_{k=0}^t \sum_{i=1}^n \lambda^{(t-k+1)p} n^{p-1}\|\boldsymbol{g}_i(\boldsymbol{x}_i^k, \boldsymbol{\xi}_i^k) - \nabla f_i(\boldsymbol{x}_i^k)\|^p\Big)^{\frac{1}{p}} \mid \mathcal{F}_{t-1}\Big]$$

$$(24)$$

$$\overset{(ii)}{\leq} 2\sqrt{2}\Big(\mathbb{E}\big[\sum_{i=1}^n \lambda^p n^{p-1}\|\boldsymbol{g}_i(\boldsymbol{x}_i^t, \boldsymbol{\xi}_i^t) - \nabla f_i(\boldsymbol{x}_i^t)\|^p \mid \mathcal{F}_{t-1}\big]$$

$$+ \sum_{k=0}^{t-1}\sum_{i=1}^n \lambda^{(t-k+1)p} n^{p-1}\|\boldsymbol{g}_i(\boldsymbol{x}_i^k, \boldsymbol{\xi}_i^k) - \nabla f_i(\boldsymbol{x}_i^k)\|^p\Big)^{\frac{1}{p}}$$

$$\overset{(iii)}{\leq} 2\sqrt{2}\Big(\lambda^p n^p \sigma^p + \sum_{k=0}^{t-1}\sum_{i=1}^n \lambda^{(t-k+1)p} n^{p-1}\|\boldsymbol{g}_i(\boldsymbol{x}_i^k, \boldsymbol{\xi}_i^k) - \nabla f_i(\boldsymbol{x}_i^k)\|^p\Big)^{\frac{1}{p}},$$

where we used relation 5 from Lemma 1 in $(i)$, Jensen's inequality in $(ii)$, and Assumption 3 in $(iii)$. From (23), taking expectations on both sides of (24), and applying the above arguments recursively from $\mathcal{F}_{t-2}$ to $\mathcal{F}_0$, we have

$$\mathbb{E}\Big[\|\sum_{k=0}^t (\boldsymbol{W} \otimes \boldsymbol{I}_d - \frac{1}{n}\boldsymbol{1}_n\boldsymbol{1}_n^\top \otimes \boldsymbol{I}_d)^{t-k+1}(\boldsymbol{g}(\boldsymbol{x}^k, \boldsymbol{\xi}^k) - \nabla F(\boldsymbol{x}^k))\|\Big] \leq 2\sqrt{2}\Big(\sum_{k=0}^t \lambda^{(t-k+1)p}\Big)^{\frac{1}{p}} n\sigma.$$

Therefore, using the relation above, and (18), (22), we reach the desired relation. $\qquad\square$

We then bound average gradient estimation errors $\epsilon_1^t = \bar{\boldsymbol{v}}^t - \overline{\nabla}F(\boldsymbol{x}^t)$.

**Lemma 7** (Average gradient estimation errors)**.** *For all $t = 0, \ldots, T$, we have*

$$\mathbb{E}\big[\|\bar{\boldsymbol{v}}^t - \overline{\nabla}F(\boldsymbol{x}^t)\|\big] \leq \beta^{t+1}\|\nabla f(\bar{\boldsymbol{x}}^0)\| + \frac{2\sqrt{2}}{n^{1-\frac{1}{p}}}\Big(\sum_{k=0}^t \beta^{(t-k)p}(1-\beta)^p\Big)^{\frac{1}{p}}\sigma + \sum_{k=0}^t \beta^{t-k+1}\Big(\frac{2\alpha\lambda}{1-\lambda} + \alpha\Big)L.$$

*Proof.* Following from the step 4 in Algorithm 1, $\forall i \in [n]$,

$$\boldsymbol{v}_i^t - \nabla f_i(\boldsymbol{x}_i^t) = \beta(\boldsymbol{v}_i^{t-1} - \nabla f_i(\boldsymbol{x}_i^{t-1})) + (1-\beta)(\boldsymbol{g}_i(\boldsymbol{x}_i^t, \boldsymbol{\xi}_i^t) - \nabla f_i(\boldsymbol{x}_i^t)) + \beta(\nabla f_i(\boldsymbol{x}_i^{t-1}) - \nabla f_i(\boldsymbol{x}_i^t)).$$
$$(25)$$

Averaging the above relation over $i = 1, \dots, n$ leads to that:

$$\boldsymbol{\epsilon}_1^t = \bar{\boldsymbol{v}}^t - \overline{\nabla}F(\boldsymbol{x}^t)$$

$$= \beta(\bar{\boldsymbol{v}}^{t-1} - \overline{\nabla}F(\boldsymbol{x}^{t-1})) + (1-\beta) \cdot \underbrace{\frac{1}{n}\sum_{i=1}^n (\boldsymbol{g}_i(\boldsymbol{x}_i^t, \boldsymbol{\xi}_i^t) - \nabla f_i(\boldsymbol{x}_i^t))}_{:=\boldsymbol{s}^t \in \mathbb{R}^d} + \beta \cdot \underbrace{\frac{1}{n}\sum_{i=1}^n (\nabla f_i(\boldsymbol{x}_i^{t-1}) - \nabla f_i(\boldsymbol{x}_i^t))}_{:=\boldsymbol{z}^t \in \mathbb{R}^d}$$

$$= \beta^{t+1}\boldsymbol{\epsilon}_1^{-1} + \sum_{k=0}^t \beta^{t-k}(1-\beta)\boldsymbol{s}^k + \sum_{k=0}^t \beta^{t-k+1}\boldsymbol{z}^k.$$

Taking Euclidean norms on both sides gives that

$$\|\boldsymbol{\epsilon}_1^t\| \le \beta^{t+1}\|\boldsymbol{\epsilon}_1^{-1}\| + \|\sum_{k=0}^t \beta^{t-k}(1-\beta)\boldsymbol{s}^k\| + \|\sum_{k=0}^t \beta^{t-k+1}\boldsymbol{z}^k\|. \tag{26}$$

We now bound the terms on the right hand side of (26) one by one. First,

$$\|\boldsymbol{\epsilon}_1^{-1}\| = \|\bar{\boldsymbol{v}}^{-1} - \frac{1}{n}\sum_{i=1}^n \nabla f_i(\bar{\boldsymbol{x}}^0)\| = \|\nabla f(\bar{\boldsymbol{x}}^0)\|. \tag{27}$$

Second, notice that $\{\beta^{t-k}(1-\beta)(\boldsymbol{g}_i(\boldsymbol{x}_i^k, \boldsymbol{\xi}_i^k) - \nabla f_i(\boldsymbol{x}_i^k))\}$ is a martingale difference sequence that falls into the pursuit of Lemma 5, and thus we obtain

$$\mathbb{E}\Big[\|\sum_{k=0}^t \beta^{t-k}(1-\beta)\boldsymbol{s}^k\|\Big]$$

$$= \frac{1}{n}\mathbb{E}\Big[\|\sum_{k=0}^t \sum_{i=1}^n \beta^{t-k}(1-\beta)(\boldsymbol{g}_i(\boldsymbol{x}_i^k, \boldsymbol{\xi}_i^k) - \nabla f_i(\boldsymbol{x}_i^k))\|\Big] \tag{28}$$

$$\le \frac{2\sqrt{2}}{n}\mathbb{E}\Big[\Big(\sum_{k=0}^t \sum_{i=1}^n \|\beta^{t-k}(1-\beta)(\boldsymbol{g}_i(\boldsymbol{x}_i^k, \boldsymbol{\xi}_i^k) - \nabla f_i(\boldsymbol{x}_i^k))\|^p\Big)^{\frac{1}{p}}\Big].$$

Note that

$$\frac{2\sqrt{2}}{n}\mathbb{E}\Big[\Big(\sum_{k=0}^t \sum_{i=1}^n \|\beta^{t-k}(1-\beta)(\boldsymbol{g}_i(\boldsymbol{x}_i^k, \boldsymbol{\xi}_i^k) - \nabla f_i(\boldsymbol{x}_i^k))\|^p\Big)^{\frac{1}{p}} \mid \mathcal{F}_{t-1}\Big]$$

$$\overset{(i)}{\le} \frac{2\sqrt{2}}{n}\Big(\mathbb{E}\Big[\sum_{k=0}^t \sum_{i=1}^n \|\beta^{t-k}(1-\beta)(\boldsymbol{g}_i(\boldsymbol{x}_i^k, \boldsymbol{\xi}_i^k) - \nabla f_i(\boldsymbol{x}_i^k))\|^p \mid \mathcal{F}_{t-1}\Big]\Big)^{\frac{1}{p}}$$

$$\le \frac{2\sqrt{2}}{n}\Big(\mathbb{E}\Big[\sum_{i=1}^n (1-\beta)^p \|(\boldsymbol{g}_i(\boldsymbol{x}_i^t, \boldsymbol{\xi}_i^t) - \nabla f_i(\boldsymbol{x}_i^t))\|^p \mid \mathcal{F}_{t-1}\Big] \tag{29}$$

$$+ \sum_{k=0}^{t-1} \sum_{i=1}^n \|\beta^{t-k}(1-\beta)(\boldsymbol{g}_i(\boldsymbol{x}_i^k, \boldsymbol{\xi}_i^k) - \nabla f_i(\boldsymbol{x}_i^k))\|^p\Big)^{\frac{1}{p}}$$

$$\overset{(ii)}{\le} \frac{2\sqrt{2}}{n}\Big(n(1-\beta)^p \sigma^p + \sum_{k=0}^{t-1} \sum_{i=1}^n \|\beta^{t-k}(1-\beta)(\boldsymbol{g}_i(\boldsymbol{x}_i^k, \boldsymbol{\xi}_i^k) - \nabla f_i(\boldsymbol{x}_i^k))\|^p\Big)^{\frac{1}{p}},$$

where we used Jensen's inequality in $(i)$ and Assumption 3 in $(ii)$. From (28), taking expectations on (29), and recursively applying the preceding arguments from $\mathcal{F}_{t-2}$ to $\mathcal{F}_0$, we have

$$\mathbb{E}\Big[\|\sum_{k=0}^t \beta^{t-k}(1-\beta)\boldsymbol{s}^k\|\Big] \le \frac{2\sqrt{2}}{n^{1-\frac{1}{p}}}\Big(\sum_{k=0}^t \beta^{(t-k)p}(1-\beta)^p\Big)^{\frac{1}{p}}\sigma. \tag{30}$$

Third,

$$\|\sum_{k=0}^{t}\beta^{t-k+1}\boldsymbol{z}^k\|$$

$$\leq \sum_{k=0}^{t}\beta^{t-k+1}\|\frac{1}{n}\sum_{i=1}^{n}(\nabla f_i(\boldsymbol{x}_i^{k-1}) - \nabla f_i(\boldsymbol{x}_i^k))\|$$

$$\leq \sum_{k=0}^{t}\beta^{t-k+1}\Big(\frac{1}{n}\sum_{i=1}^{n}\|\nabla f_i(\boldsymbol{x}_i^{k-1}) - \nabla f_i(\bar{\boldsymbol{x}}^{k-1})\| + \frac{1}{n}\sum_{i=1}^{n}\|\nabla f_i(\bar{\boldsymbol{x}}^{k-1}) - \nabla f_i(\bar{\boldsymbol{x}}^k)\|$$

$$+ \frac{1}{n}\sum_{i=1}^{n}\|\nabla f_i(\bar{\boldsymbol{x}}^k - \nabla f_i(\boldsymbol{x}_i^k)\|\Big)$$

$$\overset{(i)}{\leq} \sum_{k=0}^{t}\beta^{t-k+1}\Big(L\cdot\frac{1}{n}\sum_{i=1}^{n}\|\boldsymbol{x}_i^{k-1} - \bar{\boldsymbol{x}}^{k-1}\| + L\cdot\frac{1}{n}\sum_{i=1}^{n}\|\bar{\boldsymbol{x}}^{k-1} - \bar{\boldsymbol{x}}^k\| + L\cdot\frac{1}{n}\sum_{i=1}^{n}\|\bar{\boldsymbol{x}}^k - \boldsymbol{x}_i^k\|\Big)$$

$$\overset{(ii)}{\leq} \sum_{k=0}^{t}\beta^{t-k+1}\Big(\frac{2\alpha\lambda}{1-\lambda} + \alpha\Big)L.$$

$$(31)$$

where in $(i)$ we used Assumption 2 and in $(ii)$ we used (14) in Lemma 4. Putting relations (27)(30)(31) together leads to the final bound for this lemma. □

We next bound the stacked gradient estimation errors.

**Lemma 8** (Stacked gradient estimation errors). *For all $t = 0, \ldots, T$, we have*

$$\mathbb{E}\big[\|\boldsymbol{v}^t - \nabla F(\boldsymbol{x}^t)\|\big]$$

$$\leq \beta^{t+1}\|\nabla F(\mathbf{1}_n \otimes \bar{\boldsymbol{x}}^0)\| + 2\sqrt{2}\Big(\sum_{k=0}^{t}\beta^{(t-k)p}(1-\beta)^p\Big)^{\frac{1}{p}}n\sigma + n\sum_{k=0}^{t}\beta^{t-k+1}\Big(\frac{2\alpha\lambda}{1-\lambda} + \alpha\Big)L.$$

*Proof.* Define $\tilde{\boldsymbol{\epsilon}}_1^t := \boldsymbol{v}^t - \nabla F(\boldsymbol{x}^t) \in \mathbb{R}^{nd}$. Similar to (25), we have

$$\boldsymbol{v}^t - \nabla F(\boldsymbol{x}^t) = \beta(\boldsymbol{v}^{t-1} - \nabla F(\boldsymbol{x}^{t-1})) + (1-\beta)\underbrace{(\boldsymbol{g}(\boldsymbol{x}^t, \boldsymbol{\xi}^t) - \nabla F(\boldsymbol{x}^t))}_{:=\tilde{\boldsymbol{s}}^t \in \mathbb{R}^{nd}} + \beta\underbrace{(\nabla F(\boldsymbol{x}^{t-1}) - \nabla F(\boldsymbol{x}^t))}_{:=\tilde{\boldsymbol{z}}^t \in \mathbb{R}^{nd}}$$

$$= \beta^{t+1}\tilde{\boldsymbol{\epsilon}}_1^{-1} + \sum_{k=0}^{t}\beta^{t-k}(1-\beta)\tilde{\boldsymbol{s}}^k + \sum_{k=0}^{t}\beta^{t-k+1}\tilde{\boldsymbol{z}}^k.$$

Taking Euclidean norms on both sides gives that

$$\|\tilde{\boldsymbol{\epsilon}}_1^t\| \leq \beta^{t+1}\|\tilde{\boldsymbol{\epsilon}}_1^{-1}\| + \|\sum_{k=0}^{t}\beta^{t-k}(1-\beta)\tilde{\boldsymbol{s}}^k\| + \|\sum_{k=0}^{t}\beta^{t-k+1}\tilde{\boldsymbol{z}}^k\|.$$

Similar to the analysis in Lemma 7, we bound the right hand side above term by term. First,

$$\|\tilde{\boldsymbol{\epsilon}}_1^{-1}\| = \|\nabla F(\mathbf{1}_n \otimes \bar{\boldsymbol{x}}^0)\|.$$

Second, notice also that $\{\beta^{t-k}(1-\beta)\tilde{\boldsymbol{s}}^k\}$ is a martingale difference sequence and can be dealt with using Lemma 5. We have

$$\mathbb{E}\big[\|\sum_{k=0}^{t}\beta^{t-k}(1-\beta)\tilde{\boldsymbol{s}}^k\|\big]$$

$$= \mathbb{E}\big[\|\sum_{k=0}^{t}\beta^{t-k}(1-\beta)(\boldsymbol{g}(\boldsymbol{x}^t, \boldsymbol{\xi}^{t,b}) - \nabla F(\boldsymbol{x}^t))\|\big] \qquad (32)$$

$$\leq 2\sqrt{2}\mathbb{E}\big[\big(\sum_{k=0}^{t}\beta^{(t-k)p}(1-\beta)^p\|\boldsymbol{g}(\boldsymbol{x}^t, \boldsymbol{\xi}^t) - \nabla F(\boldsymbol{x}^t)\|^p\big)^{\frac{1}{p}}\big].$$

In addition,

$$2\sqrt{2}\mathbb{E}\big[\big(\sum_{k=0}^{t}\beta^{(t-k)p}(1-\beta)^{p}\|\boldsymbol{g}(\boldsymbol{x}^{t},\boldsymbol{\xi}^{t})-\nabla F(\boldsymbol{x}^{t})\|^{p}\big)^{\frac{1}{p}}\mid\mathcal{F}_{t-1}\big]$$

$$\overset{(i)}{\leq}2\sqrt{2}\big(\mathbb{E}\big[\sum_{k=0}^{t}\beta^{(t-k)p}(1-\beta)^{p}\|\boldsymbol{g}(\boldsymbol{x}^{t},\boldsymbol{\xi}^{t})-\nabla F(\boldsymbol{x}^{t})\|^{p}\big]\mid\mathcal{F}_{t-1}\big)^{\frac{1}{p}}$$

$$\overset{(ii)}{\leq}2\sqrt{2}\big(\mathbb{E}\big[\sum_{k=0}^{t}\beta^{(t-k)p}(1-\beta)^{p}\big(\sum_{i=1}^{n}\|\boldsymbol{g}_i(\boldsymbol{x}_i^{t},\boldsymbol{\xi}_i^{t})-\nabla f_i(\boldsymbol{x}_i^{t})\|\big)^{p}\big]\mid\mathcal{F}_{t-1}\big)^{\frac{1}{p}}$$

$$\overset{(iii)}{\leq}2\sqrt{2}\big(\mathbb{E}\big[\sum_{k=0}^{t}\sum_{i=1}^{n}\beta^{(t-k)p}(1-\beta)^{p}n^{p-1}\|\boldsymbol{g}_i(\boldsymbol{x}_i^{t},\boldsymbol{\xi}_i^{t})-\nabla f_i(\boldsymbol{x}_i^{t})\|^{p}\big]\mid\mathcal{F}_{t-1}\big)^{\frac{1}{p}} \tag{33}$$

$$=2\sqrt{2}\big(\mathbb{E}\big[\sum_{i=1}^{n}(1-\beta)^{p}n^{p-1}\|\nabla\boldsymbol{g}_i(\boldsymbol{x}_i^{t},\boldsymbol{\xi}_i^{t})-\nabla f_i(\boldsymbol{x}_i^{t})\|^{p}\mid\mathcal{F}_{t-1}\big]$$

$$+\sum_{k=0}^{t-1}\sum_{i=1}^{n}\beta^{(t-k)p}(1-\beta)^{p}n^{p-1}\|\boldsymbol{g}_i(\boldsymbol{x}_i^{t},\boldsymbol{\xi}_i^{t})-\nabla f_i(\boldsymbol{x}_i^{t})\|^{p}\big)^{\frac{1}{p}}$$

$$\leq2\sqrt{2}\big((1-\beta)^{p}n^{p}\sigma^{p}+\sum_{k=0}^{t-1}\sum_{i=1}^{n}\beta^{(t-k)p}(1-\beta)^{p}n^{p-1}\|\boldsymbol{g}_i(\boldsymbol{x}_i^{t},\boldsymbol{\xi}_i^{t})-\nabla f_i(\boldsymbol{x}_i^{t})\|^{p}\big)^{\frac{1}{p}},$$

where in $(i)$ we used Jensen's inequality, and in $(ii),(iii)$ we used relations 4,and 5 in Lemma 1, respectively. Based on (32), taking expectations on both sides of (33), and applying the above arguments from $\mathcal{F}_{t-2}$ to $\mathcal{F}_0$, we obtain

$$\mathbb{E}\big[\|\sum_{k=0}^{t}\beta^{t-k}(1-\beta)\tilde{\boldsymbol{s}}^{k}\|\big]\leq2\sqrt{2}\big(\sum_{k=0}^{t}\beta^{(t-k)p}(1-\beta)^{p}\big)^{\frac{1}{p}}n\sigma.$$

Third,

$$\|\sum_{k=0}^{t}\beta^{t-k+1}\tilde{\boldsymbol{z}}^{k}\|$$

$$\leq\sum_{k=0}^{t}\beta^{t-k+1}\|\nabla F(\boldsymbol{x}^{k-1})-\nabla F(\boldsymbol{x}^{k})\|$$

$$\leq\sum_{k=0}^{t}\sum_{i=1}^{n}\beta^{t-k+1}\|\nabla f_i(\boldsymbol{x}_i^{k-1})-\nabla f_i(\boldsymbol{x}_i^{k})\|$$

$$\leq\sum_{k=0}^{t}\sum_{i=1}^{n}\beta^{t-k+1}\big(\|\nabla f_i(\boldsymbol{x}_i^{k-1})-\nabla f_i(\bar{\boldsymbol{x}}^{k-1})\|+\|\nabla f_i(\bar{\boldsymbol{x}}^{k-1})-\nabla f_i(\bar{\boldsymbol{x}}^{k})\|+\|\nabla f_i(\boldsymbol{x}_i^{k})-f_i(\bar{\boldsymbol{x}}^{k})\|\big)$$

$$\overset{(i)}{\leq}n\sum_{k=0}^{t}\beta^{t-k+1}\big(\frac{2\alpha\lambda}{1-\lambda}+\alpha\big)L.$$

where $(i)$ follows from similar arguments in (30). □

Now we are ready to prove our main theorems.

*Proof of Theorem 1.* We observe that

$$\frac{1}{n}\sum_{t=0}^{T-1}\sum_{i=1}^{n}\mathbb{E}\big[\|\nabla f(\boldsymbol{x}_i^{t})\|\big]\leq\frac{1}{n}\sum_{t=0}^{T-1}\sum_{i=1}^{n}\mathbb{E}\big[\|\nabla f(\boldsymbol{x}_i^{t})-\nabla f(\bar{\boldsymbol{x}}^{t})\|+\|\nabla f(\bar{\boldsymbol{x}}^{t})\|\big]$$

$$\overset{(14)}{\leq}T\cdot\frac{\alpha\lambda L}{1-\lambda}+\sum_{t=0}^{T-1}\mathbb{E}\big[\|\nabla f(\bar{\boldsymbol{x}}^{t})\|\big]. \tag{34}$$

From Lemmas 3, (13), and Lemma 6,

$$\sum_{t=0}^{T-1} \alpha \|\nabla f(\bar{\boldsymbol{x}}^t)\|$$

$$\leq f(\bar{\boldsymbol{x}}^0) - f_* + \sum_{t=0}^{T-1} 2\alpha\big(\|\boldsymbol{\epsilon}_1^t\| + \|\overline{\nabla}F(\boldsymbol{x}^t) - \nabla f(\bar{\boldsymbol{x}}^t)\|\big) + \sum_{t=0}^{T-1} \frac{\alpha}{n}\sum_{i=1}^{n} \|\bar{\boldsymbol{y}}^t - \boldsymbol{y}_i^t\| + \sum_{t=0}^{T-1} \frac{L}{2}\alpha^2$$

$$\leq f(\bar{\boldsymbol{x}}^0) - f_*$$

$$+ \sum_{t=0}^{T-1} 2\alpha\Big[\beta^{t+1}\|\nabla f(\bar{\boldsymbol{x}}^0)\| + \frac{2\sqrt{2}}{n^{1-\frac{1}{p}}}\Big(\sum_{k=0}^{t}\beta^{(t-k)p}(1-\beta)^p\Big)^{\frac{1}{p}}\sigma + \sum_{k=0}^{t}\beta^{t-k+1}\Big(\frac{2\alpha\lambda}{1-\lambda} + \alpha\Big)L + \frac{\alpha\lambda L}{1-\lambda}\Big]$$

$$+ \sum_{t=0}^{T-1} \alpha\Big[2\sqrt{2}n^{\frac{1}{2}}\Big(\frac{1}{\beta} - 1\Big)\Big(\sum_{k=0}^{t}\lambda^{(t-k+1)p}\Big)^{\frac{1}{p}}\sigma + \frac{1}{\sqrt{n}}\Big(\frac{1}{\beta} - 1\Big)\sum_{k=0}^{t}\lambda^{t-k+1}\mathbb{E}\big[\|\nabla F(\boldsymbol{x}^k) - \boldsymbol{v}^k\|\big]\Big]$$

$$+ \frac{1}{2}\alpha^2 LT$$

$$\overset{(i)}{\leq} f(\bar{\boldsymbol{x}}^0) - f_* + 2\|\nabla f(\bar{\boldsymbol{x}}^0)\| \cdot \frac{\alpha}{1-\beta} + \frac{4\sqrt{2}\sigma}{n^{1-\frac{1}{p}}} \cdot \alpha(1-\beta)^{1-\frac{1}{p}}T + \frac{4L}{1-\lambda}\cdot\frac{\alpha^2 T}{1-\beta} + \frac{2L}{1-\lambda}\cdot\alpha^2 T$$

$$+ \frac{2\sqrt{2}\sigma n^{\frac{1}{2}}}{(1-\lambda)^{\frac{1}{p}}} \cdot \Big(\frac{1}{\beta} - 1\Big)\alpha T + \frac{1}{2}L\cdot\alpha^2 T$$

$$+ \frac{1}{\sqrt{n}}\Big(\frac{1}{\beta} - 1\Big)\alpha\sum_{t=0}^{T-1}\sum_{k=0}^{t}\lambda^{t-k+1}\Big(\beta^{t+1}\|\nabla F(\mathbf{1}_n\otimes\bar{\boldsymbol{x}}^0)\| + 2\sqrt{2}n\sigma(1-\beta)^{1-\frac{1}{p}} + \frac{2nL}{1-\lambda}\cdot\frac{\alpha\beta}{1-\beta}\Big)$$

where in $(i)$ we used $\beta \leq 1$, $\lambda < 1$ and Lemma 8. Denote $f(\bar{\boldsymbol{x}}^0) - f_* = \Delta_0$. Dividing $\alpha T$ from the above relation on both sides, and putting it into (34), then rearranging terms leads to

$$\frac{1}{nT}\sum_{t=0}^{T-1}\sum_{i=1}^{n}\mathbb{E}\big[\|\nabla f(\boldsymbol{x}_i^t)\|\big]$$

$$\leq \frac{\Delta_0}{\alpha T} + \frac{2\|\nabla f(\bar{\boldsymbol{x}}^0)\|}{(1-\beta)T} + 4\sqrt{2}\sigma\cdot\frac{(1-\beta)^{1-\frac{1}{p}}}{n^{1-\frac{1}{p}}} + \frac{4L}{1-\lambda}\cdot\frac{\alpha}{1-\beta} + \Big(\frac{3L}{1-\lambda} + \frac{L}{2}\Big)\alpha$$

$$+ \frac{2\sqrt{2}\sigma}{(1-\lambda)^{\frac{1}{p}}}\cdot n^{\frac{1}{2}}\Big(\frac{1}{\beta} - 1\Big) + \frac{\|\nabla F(\mathbf{1}_n\otimes\bar{\boldsymbol{x}}^0)\|}{1-\lambda}\cdot\frac{\frac{1}{\beta} - 1}{n^{\frac{1}{2}}} + \frac{2\sqrt{2}\sigma}{1-\lambda}\cdot n^{\frac{1}{2}}\Big(\frac{1}{\beta} - 1\Big)(1-\beta)^{1-\frac{1}{p}}$$

$$+ \frac{2L}{(1-\lambda)^2}\cdot n^{\frac{1}{2}}\alpha$$

$$\overset{(i)}{\leq} \frac{\Delta_0}{\alpha T} + \frac{2\|\nabla f(\bar{\boldsymbol{x}}^0)\|}{(1-\beta)T} + 4\sqrt{2}\sigma\cdot\frac{(1-\beta)^{1-\frac{1}{p}}}{n^{1-\frac{1}{p}}} + \frac{4L}{1-\lambda}\cdot\frac{\alpha}{1-\beta} + \frac{3.5L}{1-\lambda}\alpha$$

$$+ \frac{20\sqrt{2}\sigma}{(1-\lambda)^{\frac{1}{p}}}\cdot n^{\frac{1}{2}}(1-\beta) + \frac{10\|\nabla F(\mathbf{1}_n\otimes\bar{\boldsymbol{x}}^0)\|}{1-\lambda}\cdot\frac{1-\beta}{n^{\frac{1}{2}}} + \frac{20\sqrt{2}\sigma}{1-\lambda}\cdot n^{\frac{1}{2}}(1-\beta)^{2-\frac{1}{p}} + \frac{2L}{(1-\lambda)^2}\cdot n^{\frac{1}{2}}\alpha$$

$$\overset{(ii)}{\leq} O\Big(\frac{\Delta_0}{T} + \frac{2\|\nabla f(\bar{\boldsymbol{x}}^0)\|}{(1-\beta)T} + 4\sqrt{2}\sigma\cdot\frac{(1-\beta)^{1-\frac{1}{p}}}{n^{1-\frac{1}{p}}} + \sqrt{\frac{L\Delta_0}{(1-\lambda)(1-\beta)T}} + \sqrt{\frac{3.5L\Delta_0}{(1-\lambda)T}}$$

$$+ \frac{20\sqrt{2}\sigma}{(1-\lambda)^{\frac{1}{p}}}\cdot n^{\frac{1}{2}}(1-\beta) + \frac{10\|\nabla F(\mathbf{1}_n\otimes\bar{\boldsymbol{x}}^0)\|}{1-\lambda}\cdot\frac{1-\beta}{n^{\frac{1}{2}}} + \frac{\sigma}{1-\lambda}\cdot n^{\frac{1}{2}}(1-\beta)^{2-\frac{1}{p}} + \sqrt{\frac{n^{\frac{1}{2}}L\Delta_0}{(1-\lambda)^2 T}}\Big)$$

$$\overset{(iii)}{\leq} O\Big(\frac{\Delta_0}{T} + \frac{\|\nabla f(\bar{\boldsymbol{x}}^0)\|}{T^{\frac{2p-2}{3p-2}}} + \frac{\sigma}{n^{1-\frac{1}{p}}T^{\frac{p-1}{3p-2}}} + \sqrt{\frac{L\Delta_0}{(1-\lambda)T^{\frac{2p-2}{3p-2}}}} + \sqrt{\frac{3.5L\Delta_0}{(1-\lambda)T}}$$

$$+ \frac{\sigma n^{\frac{1}{2}}}{(1-\lambda)^{\frac{1}{p}}T^{\frac{p}{3p-2}}} + \frac{\|\nabla F(\mathbf{1}_n\otimes\bar{\boldsymbol{x}}^0)\|}{(1-\lambda)n^{\frac{1}{2}}T^{\frac{p}{3p-2}}} + \frac{\sigma}{1-\lambda}\frac{n^{\frac{1}{2}}}{T^{\frac{2p-1}{3p-2}}} + \sqrt{\frac{n^{\frac{1}{2}}L\Delta_0}{(1-\lambda)^2 T}}\Big)$$

$$\tag{35}$$

where in $(i)$ we take $\beta \geq 1/10$, in $(ii)$ we used

$$\alpha = \min\left(1, \sqrt{\frac{\Delta_0(1-\beta)(1-\lambda)}{4LT}}, \sqrt{\frac{\Delta_0(1-\lambda)}{3.5LT}}, \sqrt{\frac{(1-\lambda)^2\Delta_0}{2n^{\frac{1}{2}}LT}}\right), \qquad (36)$$

and in $(iii)$ we used $1 - \beta = \frac{1}{T^{\frac{p}{3p-2}}}$. $\qquad\qquad\square$

*Proof of Theorem* 2. Note that (35)(ii) still holds under the same choice of $\alpha$ in (36) and $\beta \geq 1/10$. Continuing with $1 - \beta = 1/\sqrt{T}$, we have

$$\frac{1}{nT}\sum_{t=0}^{T-1}\sum_{i=1}^{n}\mathbb{E}\left[\|\nabla f(\boldsymbol{x}_i^t)\|\right]$$

$$\leq O\left(\frac{\Delta_0}{T} + \frac{\|\nabla f(\bar{\boldsymbol{x}}^0)\|}{\sqrt{T}} + \frac{\sigma}{n^{1-\frac{1}{p}}} \cdot \frac{1}{T^{\frac{p-1}{2p}}} + \frac{1}{T^{\frac{1}{4}}}\sqrt{\frac{L\Delta_0}{1-\lambda}} + \sqrt{\frac{3.5L\Delta_0}{(1-\lambda)T}} + \frac{\sigma n^{\frac{1}{2}}}{(1-\lambda)^{\frac{1}{p}}}\frac{1}{\sqrt{T}} + \right.$$

$$\left.\frac{\|\nabla F(\mathbf{1}_n \otimes \bar{\boldsymbol{x}}^0)\|}{(1-\lambda)n^{\frac{1}{2}}} \cdot \frac{1}{\sqrt{T}} + \frac{\sigma n^{\frac{1}{2}}}{1-\lambda} \cdot \frac{1}{T^{\frac{2p-1}{2p}}} + \sqrt{\frac{n^{\frac{1}{2}}L\Delta_0}{(1-\lambda)^2 T}}\right).$$

Rearranging above terms leads to the desired upper bound. $\qquad\square$

## C   ADDITIONAL EXPERIMENT DETAILS

### C.1   BASELINE DESCRIPTIONS

Please see Table 2 for detailed descriptions of baselines.

### C.2   ADDITIONAL DETAILS FOR SYNTHETIC EXPERIMENTS

**Loss function**. Let $(\boldsymbol{X}_{i,k}, \boldsymbol{y}_{i,k})$ denote the $k$-th sample of sub-dataset $(\boldsymbol{X}_i, \boldsymbol{y}_i)$ on node $i$. The loss function of the considered nonconvex linear regression model on this sample is $\ell(\boldsymbol{y}_{i,k} - \boldsymbol{X}_{i,k}\boldsymbol{w}_i^t)$, where the

$$\ell(r) = \begin{cases} \frac{c^2}{6}\left(1 - \left[1 - \left(\frac{r}{c}\right)^2\right]^3\right) & \text{if } |r| \leq c, \\ \frac{c^2}{6} & \text{otherwise} \end{cases},$$

and we use the suggested value $c = 4.6851$ in the robust statistics literature.

**Hyperparameter tuning**. Please see Table 3 for hyperparameter searching ranges for this experiment.

**Hardware**. We ran this experiment on Mac OS X 15.3, CPU M4 10 Cores, RAM 16GB.

### C.3   ADDITIONAL DETAILS FOR DECENTRALIZED TRAINING OF TRANSFORMERS

**Transformer architecture**. We consider the following decoder-only Transformer model (GPT): vocabulary size is 10208, context length is 64, embedding size is 128, number of attention heads is 4, number of attention layers is 2, the linear projection dimension within attention block is 512, and LayerNorm is applied after the 2nd attention block. The total number of parameters of this model is 3018240.

**Hyperparameter tuning.** See Table 4 for our grid search range for algorithm hyperparameters.

**Hardware**. We simulate the distributed training on one NVIDIA H100 GPU, using PyTorch 3.2 with CUDA 12. The total hyperparameter search and training procedure took around 100 GPU hours.

Table 2: Summary of Baseline Methods

| Method | Parallel update on node $i$ | Hyper-parameters |
|---|---|---|
| DSGD | $\boldsymbol{x}_i^{t+1} = \sum_{r=1}^n w_{ir}\big(\boldsymbol{x}_r^t - \alpha g_r(\boldsymbol{x}_r^t, \boldsymbol{\xi}_r^t)\big)$ | $\alpha$: constant stepsize |
| DSGD-GClip | $\boldsymbol{x}_i^{t+1} = \sum_{r=1}^n w_{ir}\boldsymbol{x}_r^t - \alpha\,\mathsf{clip}(g_i(\boldsymbol{x}_i^t, \boldsymbol{\xi}_i^t), \tau)$ | $\alpha, \tau$: stepsize $\alpha$, and $\ell_2$ clipping levels $\tau$ |
| DSGD-CClip | $\boldsymbol{x}_i^{t+1} = \sum_{r=1}^n w_{ir}\boldsymbol{x}_r^t - \alpha\,\mathsf{clip}(g_i(\boldsymbol{x}_i^t, \boldsymbol{\xi}_i^t), \tau)$ | $\alpha, \tau$: stepsize $\alpha$, and component-wise clipping levels $\tau$ |
| DSGD-Clip | $\boldsymbol{x}_i^{t+1} = \sum_{r=1}^n w_{ir}\boldsymbol{x}_r^t - \alpha_t\,\mathsf{clip}(g_i(\boldsymbol{x}_i^t, \boldsymbol{\xi}_i^t), \tau_t)$ | $\alpha, \tau$: stepsize $\alpha_t = \alpha/(t+1)$, and $\ell_2$ clipping levels $\tau_t = \tau(t+1)^{2/5}$ |
| GT-DSGD | $\boldsymbol{y}_i^{t+1} = \sum_{r=1}^n w_{ir}\big(\boldsymbol{y}_r^t + g_r(\boldsymbol{x}_r^t, \boldsymbol{\xi}_r^t) - g_r(\boldsymbol{x}_r^{t-1}, \boldsymbol{\xi}_r^{t-1})\big)$ 
 $\boldsymbol{x}_i^{t+1} = \sum_{r=1}^n w_{ir}\big(\boldsymbol{x}_i^t - \alpha\boldsymbol{y}_r^{t+1}\big)$ | $\alpha$: constant stepsize |
| GT-Adam | $\boldsymbol{m}_i^{t+1} = \beta_1\boldsymbol{m}_i^t + (1-\beta_1)\boldsymbol{s}_i^t$ 
 $\boldsymbol{v}_i^{t+1} = \min\big(\beta_2\boldsymbol{v}_i^t + (1-\beta_2)\boldsymbol{s}_i^t \odot \boldsymbol{s}_i^t, G\big)$ 
 $\boldsymbol{x}_i^{t+1} = \sum_{r=1}^n w_{ir}\boldsymbol{x}_r^t - \alpha\frac{\boldsymbol{m}_i^{t+1}}{\sqrt{\boldsymbol{v}_i^{t+1}+\epsilon}}$ 
 $\boldsymbol{g}_i^{t+1} = \nabla f_i(\boldsymbol{x}_i^{t+1})$ 
 $\boldsymbol{s}_i^{t+1} = \sum_{r=1}^n w_{ir}\boldsymbol{s}_r^t + \boldsymbol{g}_i^{t+1} - \boldsymbol{g}_i^t$ | $\alpha, G$: constant stepsize $\alpha$, and upper bound $G$, stabilization factor $\epsilon$ |
| QG-DSGDm | $\boldsymbol{m}_i^{t+1} = \beta\hat{\boldsymbol{m}}_i^t + g_i(\boldsymbol{x}_i^t, \boldsymbol{\xi}_i^t)$ 
 $\boldsymbol{x}_i^{t+1} = \sum_{r=1}^n w_{ir}\big(\boldsymbol{x}_i^t - \eta\boldsymbol{m}_i^{t+1}\big)$ 
 $\boldsymbol{d}_i^t = (\boldsymbol{x}_i^{t+1} - \boldsymbol{x}_i^t)/\eta$ 
 $\hat{\boldsymbol{m}}_i^{t+1} = \mu\hat{\boldsymbol{m}}_i^t + (1-\mu)\boldsymbol{d}_i^t$ | $\eta, \beta, \mu$: constant stepsize $\eta$, momentum parametes $\beta, \mu$ |
| SClip-EF-Network | $\boldsymbol{m}_i^{t+1} = \beta_t\boldsymbol{m}_i^t + (1-\beta_t)\boldsymbol{\Psi}_t(g_i(\boldsymbol{x}_i^t, \boldsymbol{\xi}_i^t) - \boldsymbol{m}_i^t)$ 
 $\boldsymbol{x}_i^{t+1} = \sum_{r=1}^n w_{ir}\big(\boldsymbol{x}_r^t - \alpha_t\boldsymbol{m}_r^{t+1}\big)$ | $c_\varphi, \tau, \alpha, \beta$: Component-wise smooth clipping operator: $\boldsymbol{\Psi}_t(y) = \frac{c_\varphi}{\sqrt{t+1}}\frac{y}{\sqrt{y^2 + \tau(t+1)^{3/5}}}$, stepsize $\alpha_t = \alpha/(t+1)^{1/5}$, momentum stepsize $\beta_t = \beta/\sqrt{t+1}$. |

Table 3: Hyperparameter grid search in synthetic experiments

| Method | Hyperparameter search set |
|--------|---------------------------|
| `DSGD` | $\alpha \in \{10^{-5}, 5*10^{-5}, 10^{-4}, 5*10^{-4}, 10^{-3}, 5* 10^{-3}, 10^{-2}, 5*10^{-2}, 10^{-1}, 0.5, 1, 5, 10\}$ |
| `DSGD-Clip` | $\alpha \in \{10^{-5}, 5*10^{-5}, 10^{-4}, 5*10^{-4}, 10^{-3}, 5* 10^{-3}, 10^{-2}, 5*10^{-2}, 10^{-1}, 0.5, 1, 5, 10\}, \tau \in \{10^{-3}, 5*10^{-3}, 10^{-2}, 5* 10^{-2}, 10^{-1}, 0.5, 1, 5, 10, 50, 10^2\}$ |
| `GT-DSGD` | $\alpha \in \{10^{-5}, 5*10^{-5}, 10^{-4}, 5*10^{-4}, 10^{-3}, 5* 10^{-3}, 10^{-2}, 5*10^{-2}, 10^{-1}, 0.5, 1, 5, 10\}$ |
| `GT-NSGDm` | $\alpha \in \{10^{-5}, 5*10^{-5}, 10^{-4}, 5*10^{-4}, 10^{-3}, 5* 10^{-3}, 10^{-2}, 5*10^{-2}, 10^{-1}, 0.5, 1, 5, 10\}, \beta \in \{0.01, 0.1, 0.2, 0.3, 0.4, 0.5, 0.6, 0.7, 0.8, 0.9, 0.99\}$ |
| `SClip-EF-Network` | $\alpha \in \{10^{-3}, 10^{-2}, 0.1, 1, 10, 30\}, \beta \in \{10^{-2}, 0.1, 0.5, 0.8, 0.99\}, c_\varphi \in \{1, 5, 10, 20, 30, 50\}, \tau \in \{0.1, 1, 10, 50, 100\}$ |

Table 4: Hyperparameter grid search in decentralized training of Transformers

| Method | Hyperparameter search set |
|--------|---------------------------|
| `DSGD` | $\alpha \in \{10^{-4}, 5*10^{-4}, 10^{-3}, 5*10^{-3}, 10^{-2}, 5* 10^{-2}, 10^{-1}, 0.5, 1\}$ |
| `DSGD-GClip` | $\alpha \in \{10^{-4}, 10^{-3}, 10^{-2}, 10^{-1}, 1, 10, 10^2\}, \tau \in \{10^{-3}, 10^{-2}, 10^{-1}, 1, 10, 10^2\}$ |
| `DSGD-CClip` | $\alpha \in \{10^{-4}, 10^{-3}, 10^{-2}, 10^{-1}, 1, 10, 10^2\}, \tau \in \{10^{-3}, 10^{-2}, 10^{-1}, 1, 10, 10^2\}$ |
| `DSGD-Clip` | $\alpha \in \{10^{-4}, 10^{-3}, 10^{-2}, 10^{-1}, 1, 10, 10^2\}, \tau \in \{10^{-3}, 10^{-2}, 10^{-1}, 1, 10, 10^2\}$ |
| `GT-DSGD` | $\alpha \in \{10^{-4}, 5*10^{-4}, 10^{-3}, 5*10^{-3}, 10^{-2}, 5* 10^{-2}, 10^{-1}, 0.5, 1\}$ |
| `GT-Adam` | $\alpha \in \{5*10^{-5}, 10^{-4}, 5*10^{-4}, 10^{-3}, 5* 10^{-3}, 10^{-2}, 5*10^{-2}, 10^{-1}, 0.5, 1, 5, 10\}, G \in \{10^{-3}, 10^{-2}, 10^{-1}, 1, 10\}, \epsilon = 10^{-8}$ |
| `QG-DSGDm` | $\eta \in \{5*10^{-5}, 10^{-4}, 5*10^{-4}, 10^{-3}, 5* 10^{-3}, 10^{-2}, 5*10^{-2}, 10^{-1}, 0.5, 1, 5, 10\}, \beta = \mu \in \{0.01, 0.2, 0.4, 0.6, 0.8, 0.99\}$ |
| `GT-NSGDm` | $\alpha \in \{10^{-4}, 10^{-3}, 10^{-2}, 10^{-1}, 1, 10\}, \beta \in \{0.01, 0.2, 0.4, 0.6, 0.8, 0.99\}$ |
| `SClip-EF-Network` | $\alpha \in \{10^{-4}, 10^{-3}, 10^{-2}, 10^{-1}, 10^0, 10^1, 10^2\}, \beta \in \{0.01, 0.4, 0.8, 0.99\}, c_\varphi \in \{0.1, 1, 10, 10^2\}, \tau \in \{0.01, 0.1, 1, 10\}$ |