# OpenReview forum: "Decentralized Nonconvex Optimization under Heavy-Tailed Noise: Normalization and Optimal Convergence"
_ICLR.cc/2026/Conference — ICLR 2026 Poster_

### Official Review · Reviewer_RYNE · 2025-10-15

**Soundness:** 3
**Presentation:** 4
**Contribution:** 3
**Rating:** 8
**Confidence:** 4

**Summary:**

This paper introduces GT-NSGDm, a novel algorithm for decentralized nonconvex optimization under heavy-tailed gradient noise. By combining normalization, gradient tracking, and momentum, the method is the first to achieve an optimal convergence rate that matches the theoretical lower bound for the centralized setting. The algorithm's effectiveness is demonstrated on synthetic data and in training a transformer model.

**Strengths:**

- The paper establishes a groundbreaking optimal convergence rate for this problem setting, a significant theoretical advance.

- It addresses the well-documented issue of heavy-tailed noise in training large models and provides a robust design that works even when noise characteristics are unknown.

- The theoretical results are well-supported by comprehensive experiments on both synthetic and real-world tasks.

**Weaknesses:**

- The analysis is limited to communication graphs with "primitive and doubly stochastic weights," which may not hold in all practical scenarios.

**Questions:**

N/A

---

> ### Author Response · Authors · 2025-11-25
>
> We thank the reviewer for the encouraging feedback and recognizing the novelty of our algorithm, order-optimal rates, and practical effectiveness in both synthetic data and transformer training experiments.
>
> ##### W1: Communication graphs can be not practical in some cases
>
> > *The analysis is limited to communication graphs with "primitive and doubly stochastic weights," which may not hold in all practical scenarios.*
>
> We thank the reviewer for pointing out this critical problem for decentralized optimization. The communication topology plays a significant role in the decentralized setup.
>
> We clarify that these assumptions on weight matrices can be satisfied by a broad range of graphs and are standard in the decentralized optimization literature. Specifically, such a weight matrix can be constructed, for example, locally by metropolis weights [1] on undirected and connected graphs, on which one can construct symmetric and stochastic weight matrices that are subsumed by our assumption; see [2] and [3]. Such a weight matrix can also be constructed for some directed and strongly connected graphs that are weight-balanced; see [4].
>
> In practice, we agree that such weight matrices cannot always be easily constructed in general, for example when the network topology does not fall into the two cases we discussed above. We note that in data center applications, one can manually design the connections of working nodes and thus make such constructions possible; see, for example, [5][6]. For practical GPU clusters, communication collectives such as NCCL are often based on ring topology or tree topology [7], so implementing weight matrices under our assumption is practical.
>
> [1] Xiao, L., Boyd, S., & Lall, S. (2006). Distributed average consensus with time-varying metropolis weights. Automatica, 1, 1-4.
>
> [2] Pu, S., & Nedić, A. (2021). Distributed stochastic gradient tracking methods. Mathematical Programming, 187(1), 409-457.
>
> [3] Yuan, K., Ling, Q., & Yin, W. (2016). On the convergence of decentralized gradient descent. SIAM Journal on Optimization, 26(3), 1835-1854.
>
> [4] Xin, R., Khan, U. A., & Kar, S. (2020). Variance-reduced decentralized stochastic optimization with accelerated convergence. IEEE Transactions on Signal Processing, 68, 6255-6271.
>
> [5] Lian, X., Zhang, C., Zhang, H., Hsieh, C. J., Zhang, W., & Liu, J. (2017). Can decentralized algorithms outperform centralized algorithms? a case study for decentralized parallel stochastic gradient descent. Advances in neural information processing systems, 30.
>
> [6] Song, Z., Li, W., Jin, K., Shi, L., Yan, M., Yin, W., & Yuan, K. (2022). Communication-efficient topologies for decentralized learning with consensus rate. Advances in Neural Information Processing Systems, 35, 1073-1085.
>
> [7] Hu, Z., Shen, S., Bonato, T., Jeaugey, S., Alexander, C., Spada, E., ... & Hoefler, T. (2025). Demystifying NCCL: An in-depth analysis of GPU communication protocols and algorithms. arXiv preprint arXiv:2507.04786.

---

### Official Review · Reviewer_iF7b · 2025-10-16

**Soundness:** 3
**Presentation:** 2
**Contribution:** 3
**Rating:** 6
**Confidence:** 3

**Summary:**

The authors propose a new decentralized optimization algorithm under heavy-tailed noise that achieves the optimal convergence rate in expectation when the tail constant $p$ is known, which matches centralized lower bounds. When $p$ is unknown, they still achieve the SoTA rate. The idea of the algorithm combines gradient tracking with normalization (on the search direction) plus momentum‑based variance reduction.

**Strengths:**

1. The paper provides an algorithm achieving the optimal convergence rate $O\left(T^{-(p-1) /(3 p-2)}\right)$ for decentralized nonconvex optimization under heavy-tailed noise when $p$ is known.

2. Incorporating normalization with gradient tracking and momentum is simultaneously seeks to address all three challenges of nonlinearity, heterogeneity, and heavy-tailed noise.

3. The analysis holds for general smooth nonconvex functions, which relaxes assumptions used in some prior work.

**Weaknesses:**

1. The analysis is dense, and the presentation of the proofs may be improved for better clarify. For example, more intuitive scaffolding may be added.

2. In Table 1, why do SClip-EF-Network and DSGD-Clip perform so poorly? Have the hyperparameters been tuned properly? For the latter, if an algorithm with heavy-tailed convergence guarantees perform this poorly, then does it cast doubt on the usefulness of the theoretical bound?

**Questions:**

Please see weaknesses. For suggestions, I don't have much; the paper appears solid. One suggestion might be that for the result that training collapses under heavy-tailed noise in distributed (or non-centralized) settings, there is also the work [1] which isn't cited in the paper.

[1] Lee et al., Efficient Adaptive Federated Optimization. Arxiv, 2025.

---

> ### Author Response · Authors · 2025-11-25
>
> We thank the reviewer for the constructive feedback and for recognizing our order-optimal rates under heavy-tailed noise in decentralized setups, the effectiveness of our algorithmic design integrating normalization and gradient tracking, and the relaxation of restrictive assumptions in prior work.
>
> #### W1: Clearer proof roadmap
>
> > *The analysis is dense, and the presentation of the proofs may be improved for better clarify. For example, more intuitive scaffolding may be added.*
>
> We thank the reviewer for this suggestion to improve the presentation of our analysis. In the revised paper, we added a remark in Section 4 sketching the main proof steps for our main results (copied below) to illustrate the proof structure.
>
> The central recursion of our analysis leverages the descent lemma for
> $L$ smooth functions applied to consecutive network averages of $\{ \boldsymbol{x}_i^{t + 1} \}$ and $\{ \boldsymbol{x}_i^{t} \}$ (Lemma 3). This recursion involves two coupled error sources: consensus errors between $\{\boldsymbol{x}_i^{t}\}$, $\{\boldsymbol{y}_i^{t}\}$, and gradient estimation errors. We establish the intricate coupling between these errors through a series of intermediate lemmas. Our proof strategy proceeds as follows: we first derive two key lemmas that bound consensus errors in terms of gradient estimation errors (Lemma 4 and 6), then decompose gradient estimation errors into constituent components, including average gradient estimation errors (Lemma 7) and stack gradient estimation errors (Lemma 8),  and bound each separately. With all error bounds established, we substitute these into the main recursion and optimize hyperparameter selections to achieve the final order-optimal convergence rates.
>
>
> #### W2: Clarifications on the underperformance of two theoretically sound algorithms.
>
> > *In Table 1, why do SClip-EF-Network and DSGD-Clip perform so poorly? Have the hyperparameters been tuned properly?*
>
> Yes, we tuned these two algorithms extensively. The reviewer can refer to Table 4 in the Appendix C.5 for a detailed description of these parameter search ranges.
>
>
> > *For the latter, if an algorithm with heavy-tailed convergence guarantees perform this poorly, then does it cast doubt on the usefulness of the theoretical bound?*
>
> We want to clarify that both SClip-EF-Network and DSGD-Clip only have convergence guarantees for strongly convex functions. In addition, SClip-EF-Network assumes that gradient noise is symmetric, and DSGD-Clip assumes gradient itself is bounded. None of these assumptions are satisfied in the experiments of training Transformers. Therefore, it is possible these algorithms will fail to convergence for the tested problems. In contrast, we prove convergence of our proposed method for general non-convex function without imposing restrictive conditions on gradient noise or gradient itself, and it demonstrates strong empirical performance.
>
>
> #### Missing discussion of related work.
>
> We thank the reviewer for bringing the related work [1] to our attention, which rigorously shows that under heavy-tailed noise, distributed training can collapse with unbounded gradients, and introduces the nonlinear methods FedAda$^2$/FedAda$^2$++ to stabilize the distributed optimization process. In our revised version, we have added this missing reference with related discussions in Section 1.2 (colored in blue).
>
> [1] Lee et al., Efficient Adaptive Federated Optimization. Arxiv, 2025.

---

> > ### Comment · Reviewer_iF7b · 2025-11-26
> >
> > Thank you for the clarifications, and for adding additional scaffolding to the proof structure. My concerns are resolved. I did not point out significant issues in my original review, and therefore I will keep my positive rating.

---

### Official Review · Reviewer_E5CE · 2025-10-25

**Soundness:** 3
**Presentation:** 3
**Contribution:** 3
**Rating:** 4
**Confidence:** 3

**Summary:**

The paper studies decentralized nonconvex optimization under heavy-tailed gradient noise, assuming only the $p$-th moment is finite ($p\in(1,2]$). It proposes a method (GT-NSGDm) that combines gradient normalization, momentum-based variance reduction, and gradient tracking. Theoretically, the authors prove an order-optimal non-asymptotic rate $O\big(T^{-(p-1)/(3p-2)}\big)$ when $p$ is known (matching centralized lower bounds), and an adaptive $O\big(T^{-(p-1)/(2p)}\big)$ rate when $p$ is unknown that is topology-independent for $p<2$, yielding an $n^{1-1/p}$ speedup in the number of nodes.

**Strengths:**

- Optimal/non-asymptotic theory. Establishes $O\big(T^{-(p-1)/(3p-2)}\big)$ for decentralized nonconvex problems with heavy tails (known $p$), matching centralized lower bounds; for unknown $p$, obtains $O\big(T^{-(p-1)/(2p)}\big)$ that is independent of graph topology for $p<2$, together with an $n^{1-1/p}$ speedup.

- Design justification. A clear negative result demonstrates why gradient tracking + momentum are required beyond vanilla normalization in heterogeneous decentralized settings.

- Unified view of practical ingredients. Normalization, momentum-based variance reduction, and gradient tracking are integrated into a single algorithm with a complete analysis.

**Weaknesses:**

- Incremental feel in building blocks. Although the combination achieves new rates, the core components (normalization, gradient tracking, momentum) are mature; algorithmic novelty per se is modest.

- Assumption/tuning burden. Guarantees require primitive, doubly-stochastic mixing and nontrivial step-size/momentum schedules; several constants depend on spectral gap $\lambda$, smoothness $L$, initial gap $\Delta_0$, etc. Practical robustness when these are unknown is under-discussed.

-  Results are stated in expectation, not high-probability guarantees.

**Questions:**

The paper offers strong theory (optimal rates, topology independence for unknown $p$) and a convincing negative result, but relies on mature building blocks, has a nontrivial tuning/assumption burden, and could broaden empirical validation.

---

> ### Author Response · Authors · 2025-11-25
>
> We thank the reviewer for the constructive feedback and for recognizing the order-optimal rates, topology independent rates in some regimes, speedup analysis, and the complete analysis for our proposed algorithm with practical ingrediants and clear design justifications.
>
> #### W1:  Contributions on the basis of mature blocks.
>
> > *Incremental feel in building blocks ... algorithmic novelty per se is modest.*
>
> We appreciate the reviewer’s comments on using known core components to derive our order-optimal rates.
>
> 1. First, selecting an effective nonlinear or adaptive method to suppress heavy-tailed noise in decentralized settings was previously **unclear in the literature**. On the algorithm design, existing decentralized algorithms designed to deal with heavy-tailed noise, such as DSGD-Clip [1] and SClip-EF-Network [2], only show suboptimal convergence under restrictive conditions, i.e., bounded gradient, and symmetric noise, respectively; while in this work, our designed algorithm achieves optimal bounds without imposing such restrictive conditions.  On the analysis of existing algorithms, there is no study of, for example, decentralized gradient tracking methods under heavy-tailed noise. Both the algorithm design and the analysis are non-trivial and it requires a joint approach.
>
> 2. Second, while it is true that NSGDm achieves optimal convergence rate in centralized settings, several other techniques, such as clipping [3] and momentum variants, and combinations of clipping, normalization and momentum [4], and adaptive methods with clipping [5], also demonstrate optimal convergence rates. It was unclear which, if any, centralized methods could be adapted to the decentralized setting while preserving optimal complexity, as doing so requires more than a direct extension. In our case, we do not simply decentralize NSGDm; rather, we design a new method inspired by it. Determining how to **restructure the updates**, **where to incorporate momentum**, and most critically, how to introduce **gradient tracking into a nonlinear, noise-robust scheme** is nontrivial and requires new algorithmic ideas rather than a straightforward mapping from the centralized formulation. For example, in [6], the authors also propose to use gradient tracking + normalization, but to solve a max-margin problem. It has a different order of iterate updates and gradient tracker updates, and it uses a combine-then-adapt way to perform gradient tracking while we use a adapt-then-combine approach for gradient tracking. In terms of analysis, [6] only ensures asymptotic convergence in direction under deterministic gradients. **This example showcases that even algorithms with similar core components can have very different behavior and analyses.** As another illustrative example, used here for analogy, one may have the intuition that decentralized gradient descent and Nesterov acceleration can be trivially combined, but earlier distributed optimization work [7] shows that the placement of the Nesterov momentum term leads to drastically different algorithmic behavior. Our paper demonstrates an effective algorithmic design to handle heavy-tailed noise in the decentralized setup, in particular, under this design, we can bound the coupled gradient estimation errors and consensus errors in Lemma 6, by splitting the gradient estimation errors into average gradient estimation errors (Lemma 7), and stacked gradient estimation errors (Lemma 8).
>
> 3. Third, if we introduce some nonlinear operator such as normalization and clipping, there would be a fixed point issue as raised in Remark 3, and more seriously a non-convergence issue as shown in Claim 1. Gradient tracking and momentum are introduced to fix these issues. For decentralized methods such as decentralized SGD, gradient tracking is well-known to remove the dependence of function heterogeneity in the upper bounds. However, in this work, the benefit of gradient tracking is beyond that, since it turns a non-convergent method into an optimal method when coupled with momentum, which is not known before. This can inspire new methods where nonlinear operators are incorporated in decentralized settings, such as the clipping operator used to guarantee differential privacy.

---

> ### Author Response · Authors · 2025-11-25
>
> #### W2:  Weight matrix assumption and hyper-parameter dependence.
>
> > *Guarantees require primitive, doubly-stochastic mixing*
>
> We clarify that these assumptions on weight matrices can be satisfied by a broad range of graphs and are standard in decentralized optimization literature. Specifically, such a weight matrix can be constructed, for example, locally by metropolis weights [8], on undirected and connected graphs, on which one can construct symmetric and stochastic weight matrices satisfying our assumption, see [9] and [10]. Such a weight matrix can also be constructed for some directed and strongly connected graphs that are weight-balanced, see [11].
>
>
> > *nontrivial step-size/momentum schedules; several constants depend on spectral gap $\lambda$, smoothness $L$, initial gap $\Delta_0$, etc.*
>
>
> We try to match our hyper-parameter dependence with the centralized work [12] that uses normalized SGD with momentum to achieve order-optimal rates. We consider the hyperparameter dependence of both step-size and momentum schedules.
>
> 1. When $p$ is known, centralized setup requires knowledge of $\Delta_0, L, \sigma, T, p$, while our decentralized setup requires $\Delta_0, L, T, p, \lambda, n$. In contrast, our parameter selection is **independent of the noise moment bound constant $\sigma$**, but requires knowledge on network parameters $n, \lambda$, which are distinct to the decentralized setup.
>
> 2. When $p$ is unknown, centralized setup requires knowledge of $L, T$, while our decentralized result requires $\Delta_0, L, T, \lambda, n$. Again, decentralized case requires knowledge of $\lambda, n$, and also initial gap $\Delta_0$.
>
> > *Practical robustness when these are unknown is under-discussed.*
>
> From the above discussion, the hyperparameter dependence in decentralized setup is more complicated than centralized setups in terms of $n, \lambda$. While for optimal hyperparameter design we indeed need to know these quantities, however, as long as they belong to certain ranges that can be easily estimated, we will still achieve qualitatively similar results, thus making our approach robust to certain hyperparameters. For example, if all nodes know $n$ and set weights $W = I_n - (1/n)L$ where $L$ is the graph laplacian, one can have an estimate $\lambda \le 1 - c/n^3$ for some universal constant $c$ that can be determined. Once we have a such upper bound estimate, we can replace the upper bound terms involving $1/(1-\lambda)$ by this estimate in equation (35), then in our final proof steps, we can select a step size that is independent of $\lambda$.
>
> #### W3: High-probability bounds
>
> We thank the reviewer for raising the important point regarding high-probability guarantees. We agree that high-probability bounds provide strong guarantees regarding the tail behavior of the error. However, we respectfully argue that the Mean Squared Error (MSE) bounds presented in this work constitute a significant and necessary contribution for the following reasons:
>
> 1. Filling a critical gap: To the best of our knowledge, no prior work has established optimal rates in the mean-squared sense for decentralized optimization under heavy-tailed noise. Existing literature has left a gap regarding convergence in expectation, which our paper aims to fill. We believe designing an effective algorithm and establishing the optimal MSE rate is the foundational step before exploring tail behaviors.
>
> 2. Standard practice for new problems: Establishing convergence in expectation is a standard first step for a new problem class. By proving the MSE rates, we provide the baseline against which future work potentially focusing on tail bounds can be compared.

---

> ### Author Response · Authors · 2025-11-25
> **References**
>
> [1] Sun, C., Zhang, H., Chen, B., & Yu, L. (2025). Distributed stochastic optimization under heavy-tailed noises. IEEE Transactions on Automatic Control.
>
> [2] Yu, S., Jakovetic, D., & Kar, S. (2023). Smoothed Gradient Clipping and Error Feedback for Decentralized Optimization under Symmetric Heavy-Tailed Noise. arXiv e-prints, arXiv-2310.
>
> [3] Zhang, J., Karimireddy, S. P., Veit, A., Kim, S., Reddi, S., Kumar, S., & Sra, S. (2020). Why are adaptive methods good for attention models?. Advances in Neural Information Processing Systems, 33, 15383-15393.
>
>
> [4] Liu, Z., Zhang, J., & Zhou, Z. (2023, July). Breaking the lower bound with (little) structure: Acceleration in non-convex stochastic optimization with heavy-tailed noise. In The Thirty Sixth Annual Conference on Learning Theory (pp. 2266-2290). PMLR.
>
> [5] Chezhegov, S., Klyukin, Y., Semenov, A., Beznosikov, A., Gasnikov, A., Horváth, S., ... & Gorbunov, E. (2024). Gradient clipping improves adagrad when the noise is heavy-tailed. arXiv e-prints, arXiv-2406.
>
> [6] Taheri, H., & Thrampoulidis, C. (2023, April). On generalization of decentralized learning with separable data. In International Conference on Artificial Intelligence and Statistics (pp. 4917-4945). PMLR.
>
> [7] Jakovetić, D., Xavier, J. M. F., & Moura, J. M. (2013). Convergence rates of distributed Nesterov-like gradient methods on random networks. IEEE Transactions on Signal Processing, 62(4), 868-882.
>
> [8] Xiao, L., Boyd, S., & Lall, S. (2006). Distributed average consensus with time-varying metropolis weights. Automatica, 1, 1-4.
>
> [9] Pu, S., & Nedić, A. (2021). Distributed stochastic gradient tracking methods. Mathematical Programming, 187(1), 409-457.
>
> [10] Yuan, K., Ling, Q., & Yin, W. (2016). On the convergence of decentralized gradient descent. SIAM Journal on Optimization, 26(3), 1835-1854.
>
> [11] Xin, R., Khan, U. A., & Kar, S. (2020). Variance-reduced decentralized stochastic optimization with accelerated convergence. IEEE Transactions on Signal Processing, 68, 6255-6271.
>
> [12] Liu, Z., & Zhou, Z. Nonconvex Stochastic Optimization under Heavy-Tailed Noises: Optimal Convergence without Gradient Clipping. In The Thirteenth International Conference on Learning Representations, 2025.

---

### Official Review · Reviewer_28WH · 2025-10-31

**Soundness:** 3
**Presentation:** 2
**Contribution:** 3
**Rating:** 6
**Confidence:** 3

**Summary:**

The paper presents and analyzes a decentralized normalized SGD with momentum algorithm based on gradient tracking. Crucially, for heavy-tailed gradient noise (observed specially in transformer optimization) with known $p \in (1, 2]$ moment, the paper provides tight convergence rates in the non-convex setting. The paper also gives another bound when $p$ is unknown.

**Strengths:**

The heavy-noise gradient problem is very timely specially given the ubiquity of transformers. The decentralized optimization framework is a very well-studied  and classic setup for distributed optimization. The paper is sound and well-written for the most part. The analysis is sound, rigorous and appears to be tight as it matches the rate of centralized optimization. Experiments such as a decentralized GPT training simulation show the convergence and efficacy of the algorithm, although the performance of Clip-based algorithms is comparable.

**Weaknesses:**

-The algorithm and analysis benefits greatly from the recent works on the centralized setup (specially (Liu∗and Zhou, 2025)). The adaptations of normalized gradient descent algorithm to the decentralized setup based on gradient tracking, have been done in literature (although in a different context), including discussions on why the naive normalized Decentralized SGD fails (e.g., see section 2.4. in reference [1] below). Can the authors also please highlight the novelties in the proof needed to extend the proof of centralized setup to the decentralized setup?

-How do the hyper-parameter selections compare to the centralized setup? Is the fact that $\beta$ depends on unknown $T$ also the case in the centralized setup?

-The displayed bounds include many additive terms with complex dependencies on $n,\lambda, T,p$ and other parameters. It's not very obvious which term will dominate in practice, for example as $p$ varies.


typo:

line 589: appendix 5

[1] On Generalization of Decentralized Learning with Separable Data, Taheri et al. AISTATS 2023.

**Questions:**

please see previous part.

---

> ### Author Response · Authors · 2025-11-25
>
> We thank the reviewer for the constructive feedback and for recognizing the timeliness of the heavy-tailed setting, the relevance of the decentralized framework, and the tightness of our analysis.
>
> #### W1. Contributions beyond centralized works and prior decentralized normalized methods with gradient tracking.
>
> > *“The algorithm and analysis benefits ... the decentralized setup?”*
>
> We agree that, similar to many decentralized methods, our work benefits from the centralized setup work in dealing with heavy-tailed noise especially (Liu & Zhou, 2025). However, to achieve the same optimal time complexity under heavy-tailed noise in the decentralized setup is highly non-trivial when nonlinearity is introduced. First, in terms of algorithm design, existing decentralized algorithms designed to deal with heavy-tailed noise, such as DSGD-Clip (Sun & Chen, 2024) and SClip-EF-Network (Yu et al., 2023), only show suboptimal convergence under restrictive conditions, i.e., bounded gradient, and symmetric noise, respectively; while in this work, our designed algorithm achieves optimal bounds without imposing any such restrictive conditions. Second, in terms of analysis, there is no study of decentralized methods based on gradient tracking to handle heavy-tailed gradient noise. Third, in addition to the optimal time complexity, we manage to derive asymptotic upper bounds that achieve speedup in terms of number of nodes, i.e., $n^{1-1/p}$, which is distinctive in the decentralized setup and significant in this setup, in addition, we obtain network independent upper bounds in the leading terms, which is another contribution unique to the decentralized setup.
>
> To achieve these goals, it requires a joint approach based on effective algorithm design and new analysis. Specifically, we apply gradient tracking on local momentums and use an adapt-then-combine way to update gradient tracking terms, and apply normalization on the local gradient estimators. In our analysis, the crux lies in jointly analyzing the consensus errors and gradient estimation errors. We first bound the consensus errors with gradient estimation errors (Lemma 6). Then, we split the gradient estimation errors into average gradient estimation errors (Lemma 7), and stacked gradient estimation errors (Lemma 8). In this way, we can separately handle the gradient estimation errors under heavy-tailed noise in three separate dynamics and make the optimal rates achievable.
>
> We thank the reviewer for pointing out the related work (Taheri et al., 2023) which we missed when we prepared our paper. This works indeed presents a related algorithm design, i.e. FDLR, that utilizes gradient tracking and normalization to solve a max-margin problem, and its discussion on the non-convergence of vanilla normalized decentralized gradient descent in its Section 2.4 is also relevant. There are a few important differences compared to our work. First, in terms of algorithm design, it uses a combine-then-adapt way to update local gradient estimator and exchange the order of gradient tracking and local parameter update, which can lead to very different algorithmic dynamics (Tu & Sayed, 2012). Second, its analysis is conducted under deterministic gradient only, and the convergence is shown in the sense of direction and no rate is provided. We have incorporated related discussions into our related work Sectione 1.2 (colored in blue).
>
> **References**
>
> Liu, Z., & Zhou, Z. Nonconvex Stochastic Optimization under Heavy-Tailed Noises: Optimal Convergence without Gradient Clipping. In The Thirteenth International Conference on Learning Representations, 2025.
>
> Taheri, H., & Thrampoulidis, C. (2023, April). On generalization of decentralized learning with separable data. In International Conference on Artificial Intelligence and Statistics (pp. 4917-4945). PMLR.
>
> Tu, S. Y., & Sayed, A. H. (2012). Diffusion strategies outperform consensus strategies for distributed estimation over adaptive networks. IEEE Transactions on Signal Processing, 60(12), 6217-6234.
>
> Yu, S., Jakovetic, D., & Kar, S. (2023). Smoothed Gradient Clipping and Error Feedback for Decentralized Optimization under Symmetric Heavy-Tailed Noise. arXiv e-prints, arXiv-2310.

---

> ### Author Response · Authors · 2025-11-25
>
> #### W2. Hyperparameter selection compared to the centralized setup.
>
> > *“How do the hyper-parameter selections ... also the case in the centralized setup?”*
>
> We compare with the centralized work (Liu & Zhou, 2025) in terms of hyperparameter selections. In their hyperparameter selections, the momentum coefficient $\beta$ also depends on $T$, which is regarded as a known prior. We further compare all hyper-parameters, i.e., both step size and momentum coefficient. When $p$ is known, the centralized setup requires knowledge of $\Delta_0, L, \sigma, T$, while our decentralized setup requires $\Delta_0, L, T, \lambda, n$. In contrast, our parameter selection is independent of the noise moment bound constant $\sigma$, but requires knowledge of network parameters $n, \lambda$, which are distinct to the decentralized setup.
>
> #### W3. Dominant terms with varying parameters in particular $p$.
>
> > *The displayed bounds ... dependencies on $n,\lambda, T,p$ ... which term will dominate in practice, for example as $p$ varies.*
>
> We thank the reviewer for pointing out this important question for non-asymptotic bounds presented in this paper. In our experiments Section 5.1, we did empirical studies to verify the parameter dependence on $n, \lambda, \sigma$, some of which resonate with discussions in Remark 5 and Remark 6.
>
> When $T$ is sufficiently large, or equivalently, when the required error tolerance is small enough, the dominant terms will be $\frac{1}{T^{\frac{p-1}{3p-2}}} \cdot \Big( \sqrt{\frac{L \Delta_0}{ 1 - \lambda  } } + \frac{\sigma}{n^{1-\frac{1}{p}}} \Big)$, and $\frac{\sigma}{n^{1 - \frac{1}{p}} T^{\frac{p - 1}{2p}}}$, in Theorem 1, 2, respectively. These two terms will remain as the dominant terms for any $p \in (1, 2]$ in this regime.
>
> When $T$ is small, or equivalently, when the required error tolerance is not small enough, the dominant terms may vary with varying parameters $n, \lambda, \sigma, p$. In our revision, we added a Remark 8 to discuss the dependence in this regime, from which we copied some arguments here for the reviewer's reference. Both Theorem 1, 2 order upper bound terms in an increasing order of the rate exponent of $1/T$. To illustrate how can other parameters including $n, \lambda, \sigma, p$ change dominant terms, we can first group all terms by the rate exponents of $1/T$. In Theorem 1, we have the upper bound:
> $$
>  \frac{1}{T^{\frac{p-1}{3p-2}}} \cdot \Big( \sqrt{\frac{L \Delta_0}{ 1 - \lambda  } } + \frac{\sigma}{n^{1-\frac{1}{p}}} \Big)  +  \frac{1}{ T^{\frac{2p-2}{3p-2}}} \cdot \\| \nabla f(\bar{x}^0) \\| + \frac{1}{\sqrt{T}} \cdot \Big( \sqrt{\frac{3.5L\Delta_0}{1 - \lambda}}  + \sqrt{\frac{n^{\frac{1}{2}} L \Delta_0}{(1 - \lambda)^2}} \Big) \\  + \frac{1}{T^{\frac{p}{3p-2}} } \cdot \Big( \frac{\sigma n^{\frac{1}{2}} }{(1 - \lambda)^{\frac{1}{p}} }  +   \frac{ \\| \nabla F(\mathbf{1} \otimes \bar{x}^0)\\| }{(1 - \lambda) n^{\frac{1}{2}} } \Big)  + \frac{1}{T^{\frac{2p-1}{3p-2}}} \cdot \frac{ \sigma n^{\frac{1}{2}} }{1 - \lambda} + \frac{1}{ T} \cdot \Delta_0.
> $$
> We discuss one varying parameter while assuming other parameters fixed. First, for $1 - \lambda$, $\frac{1}{T^{\frac{p - 1}{3 p - 2}}}$ has coefficient $\frac{1}{\sqrt{1 - \lambda}}$ while the faster decaying term $\frac{1}{\sqrt{T}}$ has coefficient $\frac{1}{1 - \lambda}$. When $1 - \lambda \le \frac{n^{\frac{1}{2}}}{ T^{\frac{p}{3p- 2}} }$, the latter term $\frac{1}{\sqrt{T}} \sqrt{\frac{n^{\frac{1}{2}} L \Delta_0}{(1 - \lambda)^2}}$ will dominate. Second, for $n$, if $n^{\frac{1}{2}} \ge (1 - \lambda)T^{\frac{p}{3p-2}}$, we will also have a dominant term $\frac{1}{\sqrt{T}} \sqrt{\frac{n^{\frac{1}{2}} L \Delta_0}{(1 - \lambda)^2}}$. Third, for $p \in (1, 2]$. When $p$ goes to $1_+$, the first term will have coefficient $\sqrt{\frac{L \Delta_0}{1 - \lambda}} + \sigma$ and the speedup in terms of number of nodes will vanish, but the change in $\frac{1}{T^{\frac{p}{3p-2}}} \frac{\sigma n^{\frac{1}{2}}} {(1 - \lambda)^{\frac{1}{p}}}$ will not bring a new dominant term if we only look at dependence on $1 -\lambda$ since there already exits slower term $\frac{1}{\sqrt{T}} \frac{n^{\frac{1}{2}}}{1 - \lambda}$; Similarly , when $p$ goes $2$, the speedup in $n$ is more significant, the term $\frac{1}{T^{\frac{p}{3p-2}}} \frac{\sigma n^{\frac{1}{2}}} {(1 - \lambda)^{\frac{1}{p}}}$ only gets smaller and does not change the dominant terms.

---

> ### Author Response · Authors · 2025-11-25
>
> Although our rates are established for general $p \in (1, 2]$, we can compare parameter dependence with rates obtained for $p = 2$ only. For example, in (Xin et al., 2021), the corresponding upper bound is $$
> O\Big( \frac{\Delta_0 + \sigma \sqrt{L}}{n^{1/4} T^{1/4}} + \frac{ \sqrt{n} \sigma L }{ (1 - \lambda^2)^{3/2} \sqrt{T} } + \frac{ L \\| \nabla F(1_n \otimes \bar{x}^0) \\| }{ (1 - \lambda^2)^{3/2} T }  \Big).
> $$
> If we set $p = 2$ in our bounds, we have
> $$
> O\Big( \frac{1}{T^{1/4}} \cdot \big(\sqrt{\frac{L \Delta_0}{ 1 - \lambda  } } + \frac{\sigma}{n^{1/2}}\big) + \frac{1}{\sqrt{T}} \cdot \big( \\| \nabla f(\bar{x}^0)\\| + \sqrt{\frac{3.5L\Delta_0}{1 - \lambda}}  + \sqrt{\frac{n^{\frac{1}{2}} L \Delta_0}{(1 - \lambda)^2}} + \frac{\sigma n^{\frac{1}{2}} }{(1 - \lambda)^{ 1/2 } }  +   \frac{ \\| \nabla F(\mathbf{1} \otimes \bar{x}^0)\\| }{(1 - \lambda) n^{\frac{1}{2}} }   \big) + \frac{1}{T^{3/4}} \cdot \frac{ \sigma n^{1/2} }{ 1 - \lambda } + \frac{\Delta_0}{T} \Big).
> $$
> We compare the above two bounds. For coefficients of $1/T^{1/4}$, our bound has dependence $1/n^{1/2}$ which is faster than the $1/n^{1/4}$ in the $p = 2$ case, but the dependence in $1/\sqrt{1 - \lambda}$ is worse. For coefficients of $1/\sqrt{T}$, our bounds have dependence $1/(1 - \lambda)$ while the $p=2$ case has dependence $1/(1-\lambda^2)^{3/2}$, which is strictly larger than ours.
>
> **References**
>
> Xin, R., Khan, U. A., & Kar, S. (2021). An improved convergence analysis for decentralized online stochastic non-convex optimization. IEEE Transactions on Signal Processing, 69, 1842-1858.

---

### Author Response · Authors · 2025-12-03

Dear Area Chair,

We thank the reviewers for their constructive feedback and the area chairs for handling our submission. We received scores (confidence) of [6(3), 4(3), 6(3), 8(4)] from the reviewers, and we have carefully addressed each reviewer's comments point by point, including all concerns raised by the reviewer who assigned a score of 4(3). We have revised our paper (the rebuttal version) to incorporate the suggested changes.

The key improvements in our revised paper and additional key features of our responses include:
1. We provided a detailed analysis of hyperparameter dependencies and comparison with centralized methods, and added an example showing how the extra dependence (compared to centralized case) on the network parameter $\lambda$ can be eliminated under certain weight constructions while still obtaining order-optimal rates.
2. We conducted a careful analysis of the dependence on other problem parameters including $n$, $\lambda$, and $p$ when the required optimality gap is not small enough, i.e., when $T$ is not sufficiently large. We compare these dependencies with those of decentralized methods under classical noise (not heavy-tailed) by setting $p = 2$ in our bounds, and show that our dependence is better in terms of speedup in $n$ and in terms of $1 - \lambda$ in some cases.
3. We added a remark on the proof structure of our analysis to highlight how we bound coupled consensus errors and gradient estimation errors under heavy-tailed noise.
4. We added two missing related works on distributed optimization under heavy-tailed noise, and a method with a different construction involving normalization and gradient tracking to solve a max-margin problem.
5. We highlighted our joint approach in decentralized algorithm design and analysis to address heavy-tailed noise in an order-optimal manner. We compared our method with other constructions involving normalization and gradient tracking that lead to distinct analysis and results, and provided analogies showing how different placements of algorithmic blocks can significantly change convergence behaviors. We referred to specific technical lemmas to demonstrate why our design is effective, and discussed how it can inspire algorithm design and analysis for solving other problems, such as differential privacy, where nonlinear operators are necessary. We also highlighted our speedup analysis and network-independent rates.
6. We clarified that our assumptions on weight matrices can be satisfied by undirected and connected graphs, as well as some weight-balanced directed and strongly-connected graphs, and provided construction examples.
7. We expanded the discussion on practical implementation of weight matrices in data center and GPU cluster environments.

All new content is highlighted in blue in the revised manuscript. Below, we provide detailed point-by-point responses to each reviewer's comments.

Best regards,

Authors

---

### Meta-Review · Area_Chair_ZYt6 · 2025-12-27

**Summary:**

The paper proposes the first decentralized optimization method for non-convex problems under heavy-tailed gradient noise (with bounded $p$-th moments, $1< p \leq 2$), and establishes an in-expectation convergence rate that matches the existing lower bound for non-decentralized methods in terms of its dependence on the total number of iterations $T$. The proposed algorithm, GT-NSGDm, is built upon three key components:

- Gradient tracking, a well-known technique in decentralized optimization used to mitigate the heterogeneity of local objective functions;
- Momentum, which helps reduce the variance of the gradient estimator; and
- Normalization of updates, which is crucial for handling heavy-tailed noise.

While each of these components has been studied extensively in isolation, their integration in a decentralized, heavy-tailed setting is non-trivial. Moreover, the authors establish a $p$-agnostic convergence rate that matches the best known $p$-agnostic rate for centralized methods, as recently derived by Hubler et al. (AISTATS 2025). Numerical experiments further demonstrate that GT-NSGDm exhibits superior robustness to heavy-tailed noise compared to the other methods considered.

**Reviewer Concerns:**

Below, I summarize the reviewers’ concerns and how the authors addressed them.

**Reviewer 28WH.**

1. The reviewer asked the authors to clarify the proof novelties that are specific to the decentralized setting. The authors addressed this request appropriately in their response by clearly highlighting the decentralized-specific technical contributions.

2. The reviewer inquired about hyperparameter selection and its comparison with the centralized setting. The authors provided a clear explanation in the response, explicitly distinguishing their approach from that of Liu and Zhou (2025).

3. The reviewer noted that it was unclear which terms dominate in practice. In response, the authors added an insightful discussion and incorporated Remarks 5 and 6 into the manuscript. I believe this concern has been fully resolved.

**Reviewer E5CE.**

1. The reviewer argued that, since the individual algorithmic components are known, the algorithmic novelty is modest. I do not view this as a weakness of the paper. Many impactful results in modern research build upon existing techniques, tools, and ideas. Importantly, the authors clearly explained why their method is not a straightforward combination of prior techniques. As such, this concern has been adequately addressed.

2. The reviewer expressed concern about the dependence of hyperparameters on unknown constants. While this dependence is indeed a limitation, I do not consider it a major weakness, as the paper does not aim to develop a parameter-free method. Moreover, the authors provided a thorough discussion of these dependencies in their response.

3. The reviewer criticized the reliance on in-expectation convergence guarantees and suggested deriving high-probability bounds. I do not consider this a weakness of the current work. Both in-expectation and high-probability analyses are valuable, as the authors explained, and I fully agree with their position.

**Reviewer iF7b.**

1. The reviewer found the analysis and presentation of the proofs to be complex and suggested adding a proof sketch. The authors have addressed this by adding a proof sketch to the paper.

2. The reviewer raised concerns about the underperformance of SClip-EF-Network and DSGD-Clip. The authors provided further clarification in their response.

The reviewer subsequently acknowledged that all concerns were fully resolved.

**Reviewer RYNE.**

This review was very positive overall. The only concern pertained to the assumption of doubly stochastic weight matrices. As the authors explained, this assumption holds for a broad class of network topologies. I believe this concern has been fully resolved.

**Reviewer Scores:**

**Reviewer 28WH.** The authors have resolved all concerns raised by this reviewer. I believe the reviewer would either maintain the original score or increase it to 8.

**Reviewer E5CE.** As explained above, the issues raised by this reviewer do not constitute fundamental weaknesses of the paper. Consequently, I do not find the assigned score to be well justified. It is unclear whether the reviewer would revise their score. **Therefore, I chose not to place weight on Reviewer E5CE’s score in my final assessment.**

**Reviewer iF7b.** The reviewer acknowledged that all concerns were fully resolved and maintained the original score.

**Reviewer RYNE.** I expect this reviewer would keep the original score. The initial evaluation was very positive, the sole concern was minor, and the authors addressed it adequately.

**My assessment.** Overall, the paper makes a solid and interesting contribution. I therefore recommend acceptance.

---

### Decision · Program_Chairs · 2026-01-26

Accept (Poster)